# Fractionation of $O_2/N_2$ and $Ar/N_2$ in the Antarctic ice sheet during bubble formation and bubble-clathrate hydrate transition from precise gas measurements of the Dome Fuji ice core

*Ikumi Oyabu[1], Kenji Kawamura[1,2,3], Tsutomu Uchida[4], Shuji Fujita[1,2], Kyotaro Kitamura[1], Motohiro Hirabayashi[1], Shuji Aoki[5], Shinji Morimoto[5], Takakiyo Nakazawa[5], Jeffrey P. Severinghaus[6], Jacob D. Morgan[6]

[1] National Institute of Polar Research, Tokyo 190-8518, Japan
[2] Department of Polar Science, The Graduate University of Advanced Studies (SOKENDAI), Tokyo 190-8518, Japan
[3] Japan Agency for Marine Science and Technology (JAMSTEC), Yokosuka 237-0061, Japan
[4] Division of Applied Physics, Faculty of Engineering, Hokkaido University, Sapporo 060-8628, Japan
[5] Center for Atmospheric and Oceanic Studies, Graduate School of Science, Tohoku University, Sendai 980-8578, Japan
[6] Scripps Institution of Oceanography, University of California San Diego, La Jolla, CA 92093, USA

*Correspondence to*: Ikumi Oyabu (oyabu.ikumi@nipr.ac.jp)

**Abstract.** The variations of $\delta O_2/N_2$ and $\delta Ar/N_2$ in the Dome Fuji ice core were measured from 112 m (bubbly ice) to 2001 m (clathrate hydrate ice). Our method, combined with the low storage temperature of the samples (-50 °C), successfully excludes post-coring gas-loss fractionation signals from our data. From the bubbly ice to the middle of the bubble-clathrate transition zone (BCTZ) (112 – 800 m) and below the BCTZ (>1200 m), the $\delta O_2/N_2$ and $\delta Ar/N_2$ data exhibit orbital-scale variations similar to local summer insolation. The data in the lower BCTZ (800 – 1200 m) have large scatters, which may be caused by mm-scale inhomogeneity of air composition combined with finite sample lengths. The insolation signal originally recorded at the bubble close-off remains through the BCTZ, and the insolation signal may be reconstructed by analyzing long ice samples (more than 50 cm for the Dome Fuji core). In the clathrate hydrate zone, the scatters around the orbital-scale variability decrease with depth, indicating diffusive smoothing of $\delta O_2/N_2$ and $\delta Ar/N_2$. A simple gas diffusion model was used to reproduce the smoothing and thus constrain their permeation coefficients. The relationship between $\delta Ar/N_2$ and $\delta O_2/N_2$ is markedly different for the datasets representing bubble close-off (slope ~0.5), bubble-clathrate hydrate transformation (~1), and post-coring gas-loss (~0.2), suggesting that the contribution of the mass-independent and mass-dependent fractionation processes are different for those cases. The method and data presented here may be useful for improving the orbital dating of deep ice cores over the multiple glacial cycles and further studying non-insolation-driven signals (e.g., atmospheric composition) of these gases.

## 1 Introduction

Air trapped in polar ice sheets provides information on past climate and atmosphere. Air is transported mainly by molecular diffusion from the surface to the bottom of firn (typically 50 – 100 m below the snow surface) and trapped as air bubbles. During the bubble close-off process, relatively small gas molecules such as He, Ne, $O_2$ and Ar are preferentially excluded from

air bubbles to open pores (e.g., Bender et al., 1994a; Battle et al., 1996; Severinghaus and Battle, 2006; Huber et al., 2006). The size-dependent gas fractionation may be related to the size of channels within ice crystals (Huber et al., 2006) or the differences in the dominant diffusion mechanism (Ikeda-Fukazawa et al., 2004; Severinghaus and Battle et al., 2006). Deep

ice cores from Antarctic inland show $\delta O_2/N_2$ of -5 to -10 ‰ relative to the atmosphere due to the close-off fractionation (Bender, 2002; Suwa and Bender, 2008a, b; Kawamura et al., 2007; Extier et al., 2018). $\delta Ar/N_2$ has been rarely reported, and it is less depleted than $\delta O_2/N_2$ (Sowers et al., 1989; Bender et al., 1995; Severinghaus et al., 2009).

High correlations were found between local summer insolation and $\delta O_2/N_2$ records of the Vostok and Dome Fuji (DF) cores,

which have been used for orbital dating of the cores (Bender, 2002; Suwa and Bender, 2008a; Kawamura et al., 2007). Summer insolation may influence the physical properties of snow, which later control the close-off fractionation (Bender, 2002; Kawamura et al., 2007; Fujita et al., 2009). The phasing between the local summer insolation and $\delta O_2/N_2$ signals and its variability are typically neglected in the dating practice. However, the $\delta O_2/N_2$ data from the EDC, Vostok and DF cores around the last interglacial period show discrepancies and high-frequency variabilities, challenging the assumption of stable phasing

(Bazin et al., 2016).

The current understanding of the gas fractionation and hence the assessment of $\delta O_2/N_2$ and $\delta Ar/N_2$ as the local insolation proxy are hindered, at least partly, by the scarceness of the data that are unaffected by post-coring gas loss (during ice-core drilling and storage), which also fractionates $\delta O_2/N_2$ and $\delta Ar/N_2$ (Bender et al., 1995; Kawamura et al., 2007; Kobashi et al., 2008;

Severinghaus et al., 2009; Sowers et al., 1989). For example, configurational diffusion may occur in microcracks to produce size-dependent and weak mass-dependent fractionations (Bender et al., 1995; Huber et al., 2006). Molecular diffusion through ice matrix may also produce size-dependent fractionation (Ikeda-Fukazawa et al., 2004). The DF, Vostok and GISP2 deep cores indeed showed gradual $\delta O_2/N_2$ depletion over several years at -25 – -35 °C (Kawamura et al., 2007; Suwa and Bender, 2008a, b).


Another obstacle to understanding the close-off fractionation lies in the transformation of bubbles to clathrate hydrates. In the bubble-clathrate hydrate transition zone (BCTZ: e.g., 450 – 1200 m at Dome Fuji, (Narita et al., 1999), extreme gas fractionations on the order of several 100 ‰ can occur between individual bubbles and clathrate hydrates due to lower dissociation pressure and larger permeation coefficient of $O_2$ relative to $N_2$ (e.g., Chazallon et al., 1998; Ikeda-Fukazawa et al.,

2001). The microscopic gas fractionation, combined with the post-coring gas-loss fractionation, creates high variabilities of $\delta O_2/N_2$ and $\delta Ar/N_2$ from gas measurements in the BCTZ (Bender, 2002; Kawamura et al., 2007; Kobashi et al., 2008; Lüthi et al., 2010; Shackleton et al., 2019). Below the BCTZ, the variabilities of $\delta O_2/N_2$ and $\delta Ar/N_2$ decrease with depth over several hundred meters, possibly due to diffusive smoothing (Lüthi et al., 2010; Bereiter et al., 2014; Shackleton et al., 2019). The understanding of the smoothing process has been insufficient because of the lack of high-resolution gas data for a proper

numerical simulation (Bereiter et al., 2014). Moreover, the fact that reliable $\delta O_2/N_2$ records have been available only from the

clathrate hydrate ice (i.e., older than ~100 kyr) has hindered the discussion of how the original close-off fractionation signals are preserved through the BCTZ.

A better understanding of the gas fractionation associated with bubble close-off and clathrate hydrate formation is important for the ice-core dating and eventual reconstruction of the past atmospheric ratios. Bereiter et al. (2009) predicted that the original gas composition is preserved in the inner part of the cores for a few decades at low temperature (e.g., -50 ˚C), based on a numerical simulation of gas diffusion. The prediction was verified by analyzing the DF core, which has been stored at -50 ˚C for ~20 years, after sufficiently removing its fractionated outer part (Oyabu et al., 2020). In this work, we employ the method by Oyabu et al. (2020) and measure $\delta O_2/N_2$ and $\delta Ar/N_2$ from the bubbly ice to clathrate hydrate ice in the DF core. In addition to the discrete sampling, we obtain high-resolution, continuous gas and chemical data at selected depth intervals to understand the smoothing process of $\delta O_2/N_2$ and $\delta Ar/N_2$ below the BCTZ. Moreover, the outer ice is also measured for discussing the artifactual fractionation during storage. We analyze the data mainly based on the relationships between $\delta O_2/N_2$, $\delta Ar/N_2$ and $\delta^{18}O$ of $O_2$ to distinguish the gas fractionation processes associated with the bubble close-off, transformation of bubbles to clathrate hydrates, and post-coring gas loss. A simple gas diffusion model is also employed to constrain the permeation coefficients of $N_2$, $O_2$ and Ar, and discuss the preservation of insolation signals in the gas records through the BCTZ.

## 2 Methods

### 2.1 Ice core

We use the first Dome Fuji ice core (DF1 core) drilled from 1993 to 1997 (Watanabe et al., 2003). After transporting to Japan, the samples had been stored at −50 °C at the Institute for Low Temperature Science, Hokkaido University (until 2009) and the National Institute of Polar Research (NIPR, 2009 – present). Within the DF ice core, 105 – 450 m is bubbly ice, 450 – 1200 m is bubble-clathrate transition zone (BCTZ), and below 1200 m is bubble-free ice (Narita et al., 1999; Ohno et al., 2004).

### 2.2 Air extraction and measurements

A typical ice sample is ~60 g and ~110-mm long (corresponding to ~3 to 30 years) (Fig. 1a), which is cut out from a bulk section (~500-mm long). Because $\delta O_2/N_2$ and $\delta Ar/N_2$ near the sample surface are depleted by gas loss during storage, we removed the surface by more than 8 mm for clathrate ice (Oyabu et al., 2020) and 5 mm for bubbly ice (see below). For some samples in the BCTZ (1000 – 1200 m), we removed the surface by 20 mm (Fig. 1b) to minimize the signal of gas loss from high-pressure bubbles (Bender, 2002).

The thickness of surface removal (5 mm) for bubbly ice was determined as follows. Within a pair of samples with 3-mm removal, the sample with larger outer surface ("a2" in Fig. 1a) had generally lower $\delta O_2/N_2$ and $\delta Ar/N_2$, and slightly higher

$\delta^{18}O$ than the other sample ("a1" in Fig. 1a) (Fig. B1). No such tendency is observed with >5 mm surface removal (Fig. B1). We note that $\delta^{15}N$ is insensitive to the removal thickness (Fig. B1c), confirming its immunity from gas loss (Severinghaus et al., 2009; Oyabu et al., 2020).


The experimental procedures at NIPR are described elsewhere (Oyabu et al., 2020). Briefly, ice sample was evacuated for 2 hours and melted, and the released air was cryogenically collected into a sample tube. The air sample was split into two aliquots and measured with a mass spectrometer (Thermo DELTA V, for $\delta O_2/N_2$, $\delta Ar/N_2$, $\delta^{15}N$ and $\delta^{18}O$) and gas chromatographs (two Agilent 7890A, for $CO_2$, $CH_4$ and $N_2O$ concentrations; not shown in this study). Corrections were made for non-linearity of
the elemental and isotopic ratios upon sample pressure, and for the dependency of $\delta^{15}N$, $\delta^{18}O$ and $\delta Ar/N_2$ upon $\delta O_2/N_2$ (Bender et al., 1994b). Then, the values were normalized to the modern atmosphere (Oyabu et al., 2020). The $\delta O_2/N_2$, $\delta Ar/N_2$ and $\delta^{18}O$ values relative to the modern atmosphere were corrected for the gravitational enrichment in firn, which is nearly proportional to the mass difference between the gas pairs (Craig et al., 1988; Sowers et al., 1989). The gravitational correction can be estimated from $\delta^{15}N$ of the same sample:

$$\delta_{gravcorr} = \delta - \Delta m \times \delta^{15}N \qquad (1),$$

where $\delta_{gravcorr}$ is gravitationally corrected value, $\delta$ is measured value, and $\Delta m$ is mass difference (4 for $\delta O_2/N_2$, 12 for $\delta Ar/N_2$, and 2 for $\delta^{18}O$). We measured 522 and 162 depths with single and replicates, respectively, between 112.88 to 2001.12 m (3 to 173 kyr BP).

We tested the possibility of gas loss during the sample evacuation (Craig et al., 1988). For a bubbly ice (446.4 m) and a clathrate hydrate ice (2001.1 m), we split each sample into five thin pieces of 13 – 18 g (Fig. 1d), and evacuated them for different durations (2 pieces for 20 minutes, 1 for 2 hours, and 2 for 5 hours). There are no dependence of $\delta O_2/N_2$, $\delta Ar/N_2$, $\delta^{15}N$ and $\delta^{18}O$ on the pumping time (Fig. B2). The bubbly ice data show somewhat low $\delta O_2/N_2$ and $\delta Ar/N_2$ for the 20-min evacuation, but the result is opposite to the expectation from gas loss. Also, larger variabilities of $\delta O_2/N_2$ and $\delta Ar/N_2$ are
expected in the bubbly ice than in the clathrate hydrate ice (section 4.1). Thus, our method and core quality do not significantly fractionate the gases.

Eleven samples from 179.69 to 2496.55 m, stored for 11 years at -25 ˚C, were measured for $\delta O_2/N_2$ and $\delta^{15}N$ at Tohoku University. The analytical method was slightly modified from the previous method by Kawamura et al. (2007) as follows. The
typical sample size was 250 g, and the measurements were made with a Delta plus XP (Thermo Fisher Scientific) with 64 changeover cycles. Analytical precisions (1σ) were estimated to be 0.4 and 0.011 ‰ for $\delta O_2/N_2$ and $\delta^{15}N$, respectively.

## 2.3 High-resolution analyses

The air composition and ion concentrations were measured on five 50-cm segments at the resolutions of 1 – 2 cm. Three segments were measured for the air (1258.51 – 1258.99 m, 1389.79 – 1390.32 m and 1893.51 – 1893.99 m), and two segments were measured for the air and ions (1292.29 – 1292.82 m and 1399.03 – 1399. 48 m). In Fig. 1c, the X and Y sections were divided into 25- and 12.5-mm pieces for air and ion measurements, respectively.

For the ion measurements, the ice was decontaminated by shaving-off by ~3 mm, and then chopped off with a ceramic knife and sampled in particle-free plastic bags. They were melted in a clean laboratory (class 10,000) and analyzed for the concentrations of $Cl^-$, $SO_4^{2-}$, $NO_3^-$, $F^-$, $CH_3SO_3^-$, $Na^+$, $NH_4^+$, $K^+$, $Mg^{2+}$ and $Ca^{2+}$ using ion chromatography (Thermo Fisher Scientific Dionex ICS-5000+) (Goto-Azuma et al., 2019).

## 2.4 Diffusion model

We simulate diffusive smoothing of $\delta O_2/N_2$ and $\delta Ar/N_2$ in the clathrate hydrate ice with a one-dimensional diffusion model (Ikeda-Fukazawa et al., 2005; Bereiter et al., 2009; Bereiter et al., 2014) to test whether the observed reduction of variability below BCTZ (see 3.1.4) is consistent with molecular diffusion in ice as previously hypothesized. Because both diffusion coefficients and solubilities of $O_2$, $N_2$ and Ar in ice are poorly known, we run the model with different sets of previously proposed permeabilities (the product of diffusion coefficient and solubility).

The model assumes that the molecular diffusion through ice lattice is driven by the concentration gradient of gas molecules dissolved in ice, which are in equilibrium with clathrate hydrates in respective layers of the model. The governing equation is

$$\frac{\partial C_m^h}{\partial t} = \frac{\partial}{\partial z}\left(D_m \frac{\partial C_m^h}{\partial z}\right), \tag{2}$$

where $D_m$ is the diffusivity of $m$-molecule in ice at 1 MPa and $C_m^h$ is the concentration of $m$-molecule ($m$ = N$_2$, O$_2$ or Ar) dissolved in ice in equilibrium with clathrate hydrate (see Table A1 for the full list of symbols). $C_m^h$ is expressed as

$$C_m^h = S_m P_m^d X_m, \tag{3}$$

where $S_m$ is the solubility of $m$-molecule in ice at 1 MPa, $P_m^d$ is the dissociation pressure of the $m$-molecule, and $X_m$ is the mean molar fraction of $m$-molecule in the clathrate hydrates. The dissociation pressure (MPa) of $m$-molecule at temperature $T$ (K) is given by (Miller, 1969; Kuhs et al., 2000) (Fig. A1)

$$log P_m^d = a_m - \frac{b_m}{T}, \tag{4}$$

where $a_m$ and $b_m$ are constant and shown in Table 2.

We tested three sets of model-based permeabilities estimated by Ikeda-Fukazawa et al. (2001) (hereafter IkFk01), Salamatin et al. (2001) (hereafter Salm01), and Ikeda-Fukazawa et al. (2005) (hereafter IkFk05) (Fig. 2, Table A2, A3).

There are no published permeability of Ar in ice, thus we used two formulations proposed by Kobashi et al. (2015) (Fig. 2). The first permeability $k_{Ar(I)}$ uses diffusion coefficients of N$_2$, O$_2$ and Ar at 270 K from the molecular dynamics simulations by Ikeda-Fukazawa et al. (2004) ($D_{N_2}^{270}$: 2.1×10$^{-11}$ m$^2$ s$^{-1}$, $D_{O_2}^{270}$: 4.7×10$^{-11}$ m$^2$ s$^{-1}$ and $D_{Ar}^{270}$: 4.0×10$^{-11}$ m$^2$ s$^{-1}$):

$$k_{Ar(I)} = k_{O_2} - \left(\frac{D_{O_2}^{270} - D_{Ar}^{270}}{D_{O_2}^{270} - D_{N_2}^{270}}\right)\left(k_{O_2} - k_{N_2}\right). \quad (5)$$

The second permeability $k_{Ar(II)}$ is based on the observations that δAr/N$_2$ is often depleted about half of δO$_2$/N$_2$ (e.g.,
Severinghaus et al., 2009), and is given by:

$$k_{Ar(II)} = \frac{(k_{N_2} + k_{O_2})}{2}. \quad (6)$$

The model has the initial depth domain of 20 m, consisting of 0.5-mm-thick boxes. The initial depth profiles of the gas concentrations are given by repeating the shallowest high-resolution δO$_2$/N$_2$ and δAr/N$_2$ data (1258.51 – 1258.99 m, 81.7 kyr BP) normalized to zero-mean, converted with total air content. The model is run for 100 kyr with the timestep of ~12 days to
simulate the diffusive relaxation of the initial concentration variations and thus the composition of clathrate hydrates. To account for the actual ice sheet conditions, the box heights are gradually reduced according to the thinning function (Nakano et al., 2016) and age scale of the ice core (Kawamura et al., 2007) (Fig. A2). The temperature in the model is also changed according to the observed depth profile of borehole temperature (Buizert et al., 2021) (Fig. A2). We assume that the depth profiles of thinning and temperature are constant through time. The modeled δO$_2$/N$_2$ and δAr/N$_2$ at 2.9, 13.7, 14.5 and 67.5
kyr (modeled age) are compared with the corresponding high-resolution data at 1292, 1390, 1399 and 1894 m, respectively. Because we do not know the initial concentration profiles at the top of the clathrate-hydrate zone for each high-resolution data, it is reasonable only to compare the amplitudes and frequencies of the δO$_2$/N$_2$ and δAr/N$_2$ variations between the data and model outputs. Additional details of the model and simulation method are described in Appendix A.

## 3 Results

### 3.1 δO$_2$/N$_2$ and δAr/N$_2$ in bubbly ice, BCTZ ice and clathrate-hydrate ice

δO$_2$/N$_2$ and δAr/N$_2$ from the 11-cm samples are shown in Figure 3b and 3c. Note that all δO$_2$/N$_2$, δAr/N$_2$, and δ$^{18}$O data in this and following sections are gravitationally corrected (see 2.2). As expected, most data are negative (i.e., lower than the atmosphere) because of preferential loss of O$_2$ and Ar relative to N$_2$ during bubble close-off (Bender et al., 1995; Severinghaus and Battle, 2006). The average values of δO$_2$/N$_2$ and δAr/N$_2$ are about -10 and -5 ‰, with the overall ranges of -14 to -5 and -
7 to -3 ‰, respectively.

Variations in $\delta O_2/N_2$ and $\delta Ar/N_2$ for the depths deeper than ~1200 m have similarity with local summer insolation curve, while little similarity is found for 800 – 1200 m with extremely large scatters (Fig. 4). For the depths shallower than ~800 m, the comparisons between the gas records and insolation are less robust than for the deeper depths because of the short length (in

terms of age) and small insolation amplitudes (small signal-to-noise ratio). Nevertheless, we find similarity between $\delta O_2/N_2$ and local summer insolation in that both curves show the two peaks at ~350 m (12 kyr BP) and ~700 m (32 kyr BP) and that the second peak (at ~700 m) is larger than the first one. We assess the scatters in the data by taking residuals of $\delta O_2/N_2$ and $\delta Ar/N_2$ from their low-pass filtered curves (Fig. 3d and 3e). The low-pass filter (cut-off period: 16.7 kyr) and its usage are the same as in Kawamura et al. (2007). Briefly, we put the $\delta O_2/N_2$ and $\delta Ar/N_2$ data on the DFO-2006 time scale, linearly

interpolated them at 0.1 kyr intervals, and applied the filter to extract their orbital-scale variations. The average residuals of $\delta O_2/N_2$ are 1.1 ‰ in the bubbly ice zone (~112 – 450 m), 0.7 ‰ in the upper BCTZ (~450 – 800 m), 4.6 ‰ in the lower BCTZ (800 – 1200 m), 2.3 ‰ just below BCTZ (~1200 – 1480 m), and 0.4 ‰ in the deeper depths (~1480 – 2000 m). The average residuals of $\delta Ar/N_2$ also show similar pattern (0.5 ‰ for 112 – 450 m, 0.5 ‰ for 450 – 800m, 2.8 ‰ for 800 – 1200 m, 1.3 ‰ for 1200 – 1480 m, and 0.3 ‰ for ~1480 – 2000 m). Below, we divide our dataset into four depth ranges guided by the scatters

as well as the classic boundaries between bubbly ice zone, BCTZ and clathrate ice zone (Narita et al., 1999; Ohno et al., 2004).

### 3.1.1 112 – 450 m (bubbly ice)

The average $\delta O_2/N_2$ and $\delta Ar/N_2$ are -10.9 and -5.6 ‰, with the overall ranges of -13.0 to -7.0 and -7.2 to -3.4 ‰, respectively. Figs. 2f and 2g show differences of $\delta O_2/N_2$ and $\delta Ar/N_2$ between ice samples cut from the same depth (Fig. 1a) (hereafter

referred to as pair difference or $\Delta\delta O_2/N_2$ and $\Delta\delta Ar/N_2$). Ranges of $\Delta\delta O_2/N_2$ and $\Delta\delta Ar/N_2$ are ~0.02 to 0.97 ‰ and 0.01 to 0.96 ‰, respectively. Pair differences are usually used to calculate the pooled standard deviation, a metric of analytical precision. Instead, we use it to evaluate spatial variability in air composition for a given depth in the ice sheet (pair differences of $\delta^{15}N$ and $\delta^{18}O$ are plotted in Fig. B3 and pooled standard deviations are summarized in Table C1).

Our new $\delta O_2/N_2$ data are compared with previously obtained data at Tohoku University using the samples stored at -25 °C for various durations (Fig. 5). As expected, the samples stored at -25 °C generally show lower $\delta O_2/N_2$ than those stored at -50 °C, with dependency on the storage duration. Also, while the data from the -50 °C samples slightly increase with depth in the bubbly ice zone as expected from the insolation variation, the data from the -25 °C samples tend to decrease with depth possibly due to the increase in bubble pressure (leading to larger gas-loss) (Ikeda-Fukazawa et al., 2005; Kawamura et al., 2007).


### 3.1.2 450 – 800 m (upper BCTZ)

The number and size of clathrate hydrates gradually increase with depth, but air bubbles are still the dominant form of air inclusion (Fig. 3i, 3j) (Ohno et al., 2004). In the lowermost part of this range (below 720 m), individual clathrate hydrates with extremely enriched $\delta O_2/N_2$ (> ~1000 ‰) are found by Raman spectroscopy (Fig. 3h) (Ikeda-Fukazawa et al., 2001), in which laser light is focused on individual bubbles or clathrate hydrates, and the shift of wavelength and intensity of scattered light (Raman spectra) are measured for quantifying $O_2$ and $N_2$. The compositional ratio of $O_2$ and $N_2$ is assumed to be equal to the ratio of their Raman peak intensities. The $\delta O_2/N_2$ and $\delta Ar/N_2$ values from the gas analyses smoothly connect with those in the bubbly ice zone, with a slightly increasing trend with depth (Fig. 3b, 3c). The pair differences of $\delta O_2/N_2$ and $\delta Ar/N_2$ are smaller than 1 ‰ until ~650 m, below which they show high values (> 3 ‰, Fig. 3f, 3g).

### 3.1.3 800 – 1200 m (lower BCTZ)

Major transition from air bubbles to clathrate hydrates occur in this depth range (Fig. 3i, 3j) (Ohno et al., 2004). The $\delta O_2/N_2$ values of clathrate hydrates decrease from extremely high values towards the atmospheric value, and those of remaining air bubbles decrease to extremely negative values (~ -500 ‰) (Fig. 3h) (Ikeda-Fukazawa et al., 2001).

In contrast to the upper BCTZ, the bulk $\delta O_2/N_2$ and $\delta Ar/N_2$ in the lower BCTZ show elevated scatters (Fig. 3b-e). Very high $\delta O_2/N_2$ scatters have also been reported from the Vostok (Suwa and Bender, 2008a), GISP2 (Kobashi et al., 2008) and WAIS Divide (Shackleton et al., 2019) ice cores, with numerous positive values (higher than the atmosphere). The previous data were affected by the post-coring gas loss, thus they were interpreted only as artifacts created by the preferential gas loss from extremely $N_2$-rich air bubbles (Bender, 2002; Ikeda-Fukazawa et al., 2001; Kobashi et al., 2008; Shackleton et al., 2019). In our data, relatively few samples (six and ten samples for $\delta O_2/N_2$ and $\delta Ar/N_2$, respectively) show positive values, and even the samples with twice the surface removal (20 mm) show large scatters. Thus, the large $\delta O_2/N_2$ and $\delta Ar/N_2$ variabilities naturally occur in this zone. The pair differences of $\delta O_2/N_2$ and $\delta Ar/N_2$ are also large (up to ~10 ‰, Fig. 3g, 3h; Table C1), indicating large inhomogeneities also in the horizontal direction.

### 3.1.4 1200 – 1980 m (clathrate hydrate ice)

Between 1200 and 1480 m, the $\delta O_2/N_2$ and $\delta Ar/N_2$ data show smaller scatters than those in the lower BCTZ (Fig. 3b-e), and their low-pass filtered curves (corresponding to ~80 to 100 kyr BP) show similarity to the local summer insolation (Fig. 4). The residuals of the data from the filtered curves are up to ~5 and ~3 ‰ for $\delta O_2/N_2$ and $\delta Ar/N_2$, respectively. The large scatters below the BCTZ have also been found in the WAIS Divide (Shackleton et al., 2019), EDML and EDC (Lüthi et al., 2010) ice

cores. The pair differences of $\delta O_2/N_2$ and $\delta Ar/N_2$ are $1 - 2$ ‰ just below the BCTZ (~1200 m), and they decrease with depth to < 0.1 ‰ (Fig. 3f, 3g, Table C1).

Below 1480 m, the $\delta O_2/N_2$ and $\delta Ar/N_2$ variations show remarkable similarities with the local summer insolation with small scatters (Fig. 4). The pair differences are also small (Fig. 3f, 3g, Table C1). The previous $\delta O_2/N_2$ data corrected for the gas-loss fractionation (measured at Tohoku University; Kawamura et al., 2007) agree well with our new data (Fig. 5), suggesting that the inner part of the ice core stored at -50 °C has kept the original $\delta O_2/N_2$ for two decades.

### 3.2 Outer ice

In almost all outer ice samples, $\delta O_2/N_2$ and $\delta Ar/N_2$ are significantly lower and $\delta^{18}O$ is significantly higher than those in the inner ice, as expected for gas loss (Ikeda-Fukazawa et al., 2005; Severinghaus et al., 2009). The outer ice stored at -30 ˚C for ~1 year shows larger gas-loss signals than those stored at -50 ˚C (Fig. B4), indicating that the gas-loss fractionation strongly depends on the storage temperature and duration. In the lower BCTZ, some $\delta O_2/N_2$ and $\delta Ar/N_2$ values from the outer ice are positive (Fig. B4a, B4b), possibly due to the preferential gas loss from $N_2$-rich bubbles (Bender et al., 1995). We do not find significant differences between the outer and inner pieces for $\delta^{15}N$, $CH_4$ concentration and $N_2O$ concentration (Fig. B4d – f). The results are consistent with the earlier notion that only the molecules smaller than 3.6 Å in collision diameter ($O_2$ and Ar among the above species) significantly fractionate (Huber et al., 2006; Severinghaus and Battle, 2006).

### 3.3 High-resolution data

The high-resolution data (~1258, 1292, 1390, 1399 and 1894 m) are shown in Figure 6. For the upper four depths, $\delta O_2/N_2$ and $\delta Ar/N_2$ fluctuate with very large amplitudes (peak-to-peak: $10 - 30$ ‰) and the cycles of $10 - 15$ cm. In the deepest sample (1894 m), the values are quite stable. The standard deviations of $\delta O_2/N_2$ are 8.4, 5.9, 5.8, 3.1 and 0.2‰, and those of $\delta Ar/N_2$ are 12.5, 6.4, 5.6, 3.4 and 0.2 ‰ for 1258, 1292, 1390, 1399 and 1894 m, respectively.

The data are averaged over 4 to 5 consecutive samples and compared with the data from the normal (11 cm) samples (Fig. 6). Two normal samples at ~1293 and 1390 m are taken from the same depths as the high-resolution measurements, and they agree with the averages of the high-resolution data. The standard deviation of the averaged data (blue curves in Fig. 6) for the upper four depths are 1.4 to 2.8 ‰ for $\delta O_2/N_2$ and 0.8 to 1.5 ‰ for $\delta Ar/N_2$, which are significantly higher than that for the deepest sample (0.1 and 0.2 ‰). High variabilities at 10-cm scales have also been found in the continuous measurements of the GRIP core (in the BCTZ, Huber and Leuenberger, 2004) and EDML core (just below the BCTZ, Lüthi et al., 2010). Also,

in the GRIP data, the variabilities of $\delta O_2/N_2$ well below the BCTZ (~2500-m depth) are much smaller than those in the lower BCTZ (~1100 – 1400 m) (Huber and Leuenberger, 2004).

Major ion concentrations ($Na^+$, $Mg^{2+}$, $Ca^{2+}$, $Cl^-$, $SO_4^{2-}$ and $NO_3^-$) at 12.5-mm resolution from the two depths (1258 and 1399 m) are shown in Figure 7. The $Na^+$, $Mg^{2+}$ and $Ca^{2+}$ concentrations show similar variations, and they appear to be correlated with the gas records. For example, narrow dips in the ion concentrations at 1258.88 – 1258.89 and 1258.94 – 1258.95 m possibly correspond to the dips in the gas records. Another example is common dips in the $Na^+$, $Mg^{2+}$, $Ca^{2+}$, $Cl^-$ and $SO_4^{2-}$ concentrations and the gas data for 1399.17 – 1399.18 m. On the other hand, the $Cl^-$, $SO_4^{2-}$ and $NO_3^-$ concentrations show smooth variations possibly due to their higher mobility in firn and ice. The correlation coefficients between the ion

concentrations and gas ratios are summarized in Table 1. For calculating the correlation coefficients, the ion data are resampled to the depths of the gas data. Significant correlations with the gas data are found for the $Na^+$, $Mg^{2+}$ and $Ca^{2+}$ concentrations.

### 3.4 Diffusion model

The results of the diffusion model for 2.9, 13.7, 14.5 and 67.5 kyr (at 1292, 1390, 1399 and 1894 m, respectively) after the
initial state (81.7 kyr BP at 1258 m, Fig. 6) are resampled at 2.5-cm intervals and compared with the data (Fig. 8). The IkFk05 permeation parameters give the smoothest $\delta O_2/N_2$ profiles with a poor agreement with the data (Fig. 8c). The IkFk01 parameters produce the profiles similar to the data at 1292 m, but the model results are too smooth at 1390 and 1399 m (Fig. 8b). The results with the Salm01 parameters agree well with the data, including rapid changes (>10 ‰) within a few consecutive samples and the standard deviation (~3 ‰) at 1399 m (Fig. 8a). We note that the data at 1390 and 1399 m show
rather different standard deviations (5.8 and 3.1 ‰) probably reflecting the original fractionations in the BCTZ. Thus, the model-data comparison is inadequate with the small number of cases. For Ar, the model results with the scaling function Ar(II) are closer to the data than those with Ar(I) (Fig. 9).

### 4 Discussion

### 4.1 Fractionation in firn

The $\delta O_2/N_2$ and $\delta Ar/N_2$ data below BCTZ show variations similar to the local summer insolation (Fig. 4). In addition, we find the possible insolation signals in the bubbly ice zone and upper BCTZ (see 3.1), as expected from the proposed link between the local summer insolation and close-off fractionation through the effects on the snow metamorphism (Bender, 2002; Fujita et al., 2009).


In the bubbly ice zone, the scatters of the data are significantly larger than those of the pair differences, probably reflecting high variabilities of natural close-off fractionation in the vertical direction. In the Dome Fuji region, firn layerings of several 10 cm are found in $\delta^{18}O$ of ice, ion concentrations and density (Hoshina et al., 2014; Fujita et al., 2016). The layers closed-off deeper may be less depleted in $O_2$ and Ar because (1) air bubbles spend a shorter time in the close-off zone, and (2) $\delta O_2/N_2$

and $\delta Ar/N_2$ are enriched deeper in the open pores (e.g., Severinghaus and Battle, 2006). Also, microbubbles may form near the surface (Lipenkov, 2000) and become extremely depleted in $O_2$ and Ar (Ohno et al., 2021), also causing the $\delta O_2/N_2$ and $\delta Ar/N_2$ variations (Kobashi et al., 2015). Hereafter, we collectively call the natural fractionation during the firn densification and bubble formation as "close-off fractionation".

As discussed above, the relationship between $\delta Ar/N_2$ and $\delta O_2/N_2$ in our data is expected only to reflect natural fractionations. For bubbly ice, we find a high correlation between $\delta Ar/N_2$ and $\delta O_2/N_2$ with the slope of 0.50±0.01 (Fig. 10c, Table C2), which agrees with that of pair differences (0.53±0.04) (Fig. 10d, Table C2). The smaller fractionation for $\delta Ar/N_2$ is consistent with the size-dependent fractionation. We find similar slopes in the upper BCTZ, lower BCTZ and below the BCTZ (0.45±0.01, 0.61±0.01, 0.42±0.02, respectively, Fig. 10e, 10g, 10i, Table C2), despite large scatters in the lower BCTZ and just below the

BCTZ (~800 – 1480 m). We interpret the similarity of the slopes as preservation of the close-off fractionation signals through the BCTZ. The slight differences between the slopes in the different zones might arise from different temperature or accumulation rate in the past (affecting the close-off fractionation), or natural variations of $O_2$ in the atmosphere.

A firn-air study at WAIS Divide (Battle et al., 2011) provided the evidence of mass-dependent fractionation associated with

the close-off process with the slope of $\delta^{18}O$ vs. $\delta O_2/N_2$ of -0.0090 (‰ / ‰). For the DF ice-core data, the pair differences in the bubbly ice zone (Fig. B5), as well as the data for the last 2000 years (Oyabu et al., 2020) do not show significant correlations between $\delta^{18}O$ and $\delta O_2/N_2$. Thus, our data cannot verify the mass-dependent fractionation, and much higher precision is required to detect the $\delta^{18}O$ change of ~ 0.01 ‰ expected for the range of DF $\delta O_2/N_2$.

From the comparison of three Antarctic inland cores (EDC, Vostok, DF) that showed large short-term $\delta O_2/N_2$ variabilities and discrepancies between the cores around the last interglacial period, Bazin et al. (2016) discussed the possibility of non-orbital-scale variabilities of close-off fractionation. In our data, the residuals of $\delta O_2/N_2$ from the filtered curve are small ($1\sigma$ = ~0.5 ‰) below ~1480 m (Fig. 3d), whereas the previous data (after gas-loss correction) show the variability of 1.3 ‰ for the similar depth range (Kawamura et al., 2007) (Fig. 5). Thus, the large short-term variabilities in the previous datasets may be mostly

attributed to the poor ice core quality, and the actual short-term variability of close-off fractionation may be rather small, at least for the DF core.

## 4.2 Bubble-clathrate transformation fractionation

In the upper BCTZ, the residuals of $\delta O_2/N_2$ and $\delta Ar/N_2$ from the low-pass filtered curves exhibit scatters of ~3 ‰ and ~2 ‰
(peak-to-peak), respectively, which are much smaller than those in the lower BCTZ (Fig. 4). Pair differences are similar to
those at the bottom of the bubbly ice zone, suggesting insignificant effects of the clathrate hydrate formation on the measured
$\delta O_2/N_2$ and $\delta Ar/N_2$ until ~40 % of air bubbles are transformed to clathrate hydrates. This may be reasonably explained by the
dominance of direct conversion of air bubbles to clathrate hydrates (thus little displacements of molecules) in the early stages
of bubble-clathrate transformation (Lipenkov, 2000). It may also be the case that the distance of gas diffusion from the bubbles
to clathrate hydrates, even if it occurs, is too short to create scatters in the bulk $\delta O_2/N_2$ and $\delta Ar/N_2$.

In the lower BCTZ, the scatters around the orbital scale variations dramatically increase for $\delta O_2/N_2$ and $\delta Ar/N_2$ despite the
sufficient removal of the outer ice (Fig. 3, 4). Microscopic observations by Ohno et al. (2004) found layered distributions of
air bubbles and clathrate hydrates in the lower BCTZ, as well as high spatial variability of total number of air inclusions on a
few mm-scales, possibly due to diminishing bubbles by transferring their molecules to nearby clathrates. Using their number
concentrations of air bubbles and clathrate hydrates, the average distance between air inclusions is estimated to be 1.1 mm.
Thus, the distance of air migration associated with the bubble-clathrate transformation should be on this order. Ikeda-Fukazawa
et al. (2001) observed extremely fractionated bubbles and clathrate hydrates in the lower BCTZ, up to about +1000 ‰ for
clathrate hydrates and -740 ‰ for bubbles for $\delta O_2/N_2$. From these observations, we suggest that the highly fractionated bubbles
and clathrate hydrates may be stratified in mm-scale layers, and that the scatters in our dataset may be produced by random
inclusion of such fractionated layers at the top and/or bottom of the ice samples. For example, if a 100-mm-long ice sample
coincidentally includes a 1-mm-thick anomalous layer with $\delta O_2/N_2$ of +1000 ‰, the $\delta O_2/N_2$ of the bulk sample should be
elevated by ~10 ‰ relative to the value without the anomalous layer. We indeed observe the residual $\delta O_2/N_2$ of up to ~10 ‰
in the lower BCTZ around the orbital-scale fitting curve. Thus, by simply analyzing longer samples, the scatters created by the
thin anomalous layers should be reduced. We suggest that a sufficient sample length to reduce the scatter to an acceptable level
is ~50 cm, which would produce anomalies of up to ~2 ‰. With this noise level in the $\delta O_2/N_2$ data, the insolation signal should
be reconstructed in the BCTZ, as seen in the somewhat scattered depths just below the BCTZ (1200 – 1480 m). We also
emphasize that the removal of the gas-loss fractionated outer ice is a prerequisite for the practice of averaging longer samples,
for better reconstruction of average $\delta O_2/N_2$ in the ice sheet.


The slopes of $\delta Ar/N_2$ vs. $\delta O_2/N_2$ from the pair differences are close to 1 for both the upper and lower BCTZ (Fig. 10f and 10h,
Table C2). They are strikingly different from the that in the bubbly ice (Fig. 10d, Table C2) and should be created by the
fractionations associated with bubble-clathrate transition, because the amplitudes of $\Delta \delta O_2/N_2$ and $\Delta \delta Ar/N_2$ are larger than
those in the bubbly ice by an order of magnitude. We also investigate the pair differences of $\delta^{18}O$ vs. $\delta O_2/N_2$, which may be
an indicator of mass-dependent fractionation. We find no correlation between $\Delta \delta^{18}O$ and $\Delta \delta O_2/N_2$ with relatively wide ranges

of $\Delta\delta O_2/N_2$ (up to 10 ‰, Fig. B5), indicating that the fractionation by the bubble-clathrate transformation is not mass-dependent. This is clearly different from the known correlations between $\Delta\delta^{18}O$ and $\Delta\delta O_2/N_2$ for bubble close-off (Battle et al., 2011) and post-coring gas loss (Severinghaus et al., 2009). Thus, we suggest that the net fractionation associated with the bubble-clathrate transformation and molecular diffusion between air inclusions within the ice sheet is mass-independent, and that the bubble close-off and post-coring gas-loss fractionations include mass-dependent processes for molecules smaller than 3.6 Å such as He, $O_2$ and Ar (Craig and Scarsi, 1997; Severinghaus et al., 2009) (Fig. 11). For the BCTZ, the mass fluxes of gases from bubbles to clathrates through ice may depend on permeation coefficient and dissociation pressure (Eq. 8 in Salamatin et al., 2001), with larger permeation coefficient and lower dissociation pressure leading to larger flux. Thus, for the case of Ar and $O_2$, the lower permeation coefficient of Ar than that of $O_2$ ($2\times10^{-20}$ and $3\times10^{-20}$ $m^2$ $s^{-1}$ $MPa^{-1}$ at 240K, respectively) may be counteracted by the lower dissociation pressure of Ar than $O_2$ (3.5 and 4.9 MPa at 240K, respectively), to result in similar relative fractionation between bubbles and clathrates with respect to $N_2$. This hypothesis may explain the observed similarity of $\delta Ar/N_2$ and $\delta O_2/N_2$ enrichment in clathrates. For the bubble close-off, such a cancellation cannot occur, explaining the observed $\delta Ar/N_2$ that is only half as enriched as $\delta O_2/N_2$. The bubble close-off fractionation of small molecules (<3.6 Å molecular diameter) appears to be caused by permeation through the thin ice wall of freshly-formed bubbles, which create both size-dependent and mass-dependent fractionations ; Ikeda-Fukazawa et al., 2004; Severinghaus and Battle, 2006; Battle et al., 2011).

We find similar slopes (~1) of $\Delta\delta Ar/N_2$ vs. $\Delta\delta O_2/N_2$ in the BCTZ of the WAIS Divide (Seltzer et al., 2017) and South Pole (Severinghaus, 2019) ice cores, although the samples have possibly experienced post-coring gas loss to some extent (Table C2). The fact that the different cores from a wide range of temperature show similar relationships between $\Delta\delta Ar/N_2$ and $\Delta\delta O_2/N_2$ may suggest that the permeation coefficients of $O_2$ and Ar in ice have similar dependence on temperature.

### 4.3 Preservation of insolation signal through BCTZ

Below the BCTZ, the low-pass filtered $\delta O_2/N_2$ and $\delta Ar/N_2$ show close resemblance to the variations of local summer insolation, although the residuals are rather scattered until 1480 m (Fig. 3, 4). The large scatters probably originate in the extreme layered fractionation in the lower BCTZ. Until the conversion of bubbles to clathrate hydrates complete (at ~1200 m), gas flux from the remaining (extremely fractionated) bubbles to clathrate hydrates should continue and enhance the inhomogeneity of gas composition. Once all bubbles disappear, the gas flux occurs only between clathrate hydrates; thus, their compositions become gradually homogenized. The major homogenization finishes within 300 m (~25 kyrs) below the BCTZ, and further homogenization continues to ~500 m (~50 kyrs) below the BCTZ where the scatters become small and stable (±0.2 ‰) (Fig. 3d, 3e and 12).

In contrast to the scatters from the 11-cm samples discussed above, the pair differences decrease sharply at the bottom of BCTZ (Fig. 3f, 3g), suggesting smaller spatial variability of gas composition in the horizontal direction. The 2.5-cm continuous data from the four 50-cm samples (1258 to 1399 m) also show much higher variabilities (5 – 10 ‰ difference between neighboring samples) than the pair differences (Fig. 3f, 3g and 6). Thus, our data consistently show the large anisotropy of gas fractionation, as expected from the layered clathrate hydrate formation in the BCTZ (Ohno et al., 2004). The positive correlations between the gas ratios and $Na^+$, $Mg^{2+}$ and $Ca^{2+}$ concentrations are consistent with the suggestion by Ohno et al. (2010) that nucleation of clathrate hydrates tends to occur in the layers with abundant micro-inclusions, because the elements Na, Mg and Ca likely exist as water-soluble salts (Oyabu et al., 2014 and references therein). Thus, the high-micro-inclusion layers may create early clathrate nucleation, which attract $O_2$ and Ar from air bubbles in the adjacent layers with fewer micro-inclusions, and increase their $\delta O_2/N_2$ and $\delta Ar/N_2$. The same conclusion was reached by Huber and Leuenberger (2004) (albeit with ~50 times larger measurement uncertainties than ours), based on the quasi-annual cycles in the $\delta O_2/N_2$ and $\delta Ar/N_2$ in the GRIP ice core, Greenland, and their similarity with $Ca^{2+}$ concentrations.

The molecular diffusivities should depend on the direction in the ice sheet and also contribute to the horizontal homogeneity. The c-axes orientation gradually becomes clustered around the vertical, and the crystal grains become horizontally elongated, by the vertical compression within the ice sheet (Azuma et al., 2000). In terms of molecular diffusion through ice matrix, the diffusivity in the direction perpendicular to the c-axis is greater than that in the direction parallel to the c-axis (Ikeda-Fukazawa et al., 2004). For the diffusion along grain boundaries, the pathway should be shorter in the horizontal direction due to the crystal elongation. In addition, lattice defect and dislocation within ice crystals tend to develop along the basal plane (Hondoh, 2009), possibly providing another pathway for gas diffusion. All these mechanisms should contribute to preferential homogenization of gases in the horizontal direction.

This study, for the first time, directly compares the diffusion model results with the detailed $\delta O_2/N_2$ and $\delta Ar/N_2$ in the ice sheet. The model could reproduce the smoothing of layered gas compositions as seen in the high-resolution continuous data (Fig. 8a and 9b). Also, the relationships between $\delta Ar/N_2$ and $\delta O_2/N_2$ in different zones (bubbles, BCTZ and clathrates) are similar to each other (slope of around 0.5) (Fig. 10). From these observations, we conclude that the large scatters just below the BCTZ originate in layered gas fractionations in the lower BCTZ, and that the subsequent decrease of scatters is due to diffusive homogenization (Fig. 11). Thus, we favor the possibility that the recording mechanism of insolation variations at Dome Fuji was intact when the layers in the modern BCTZ and below were initially formed at the past firn-ice transition.

We also analyze our data in a similar manner as the work by Bereiter et al. (2014). The standard deviations of the model results resampled at 11-cm intervals are compared with the residual $\delta O_2/N_2$ and $\delta Ar/N_2$ data from the low-pass filtered curves (Fig. 12) (following Bereiter et al., 2014). Exponential fitting curves through the residual data (black line) are in close agreements with the model results with the Salm01 and Ar(II) permeation parameters. On the other hand, the model results with the other

parameters (IkFk01, IkFk05 and Ar(I)) show too rapid decrease of scatters in comparison with the data. Therefore, our datasets (high-resolution and normal datasets) consistently support the Salm01 and Ar(II) permeation parameters at around 240 K (temperature at DF for the simulated depths).


From the Salm01 parameters, the rate of diffusive migration is on the order of 0.1 mm per 10 kyr ($10^{-10}$ m s$^{-1}$). Therefore, we favor the interpretation that the extreme scatter of $\delta O_2/N_2$ and $\delta Ar/N_2$ in the BCTZ in our datasets are caused by mm-scale inhomogeneity of the compositions of air inclusions combined with the finite sample length, rather than by cm-scale bulk migration of gas molecules. We also suggest that the original insolation signal on $\delta O_2/N_2$ and $\delta Ar/N_2$ in the BCTZ may be

reconstructed by analyzing long ice samples (>50 cm) to average out the inhomogeneity (see Section 4.2).

### 4.4 Gas-loss fractionation

In the outer ice, the $\delta O_2/N_2$ and $\delta Ar/N_2$ are significantly depleted and $\delta^{18}O$ is enriched due to post-coring gas loss, with higher storage temperature leading to larger fractionation (Fig. B4). The slope of $\Delta\delta Ar/N_2$ vs. $\Delta\delta O_2/N_2$ is 0.22 (Fig. 10l, Table C2),

which is significantly smaller than those from the pairs of inner ice, suggesting that the process for gas loss is different from natural fractionation processes within the ice sheet. The slope of $\Delta\delta^{18}O$ vs. $\Delta\delta O_2/N_2$ is -0.0083 (Fig. B5), which is similar to those previously reported for artifactual gas loss from the Siple Dome (Severinghaus et al., 2009), WAIS Divide (Seltzer et al., 2017) and EDC (Extier et al., 2018) ice cores.

Severinghaus et al. (2009) extensively measured the Siple Dome ice core and found a tight relationship between $\Delta\delta Ar/N_2$ and $\Delta\delta O_2/N_2$ with the slope of ~0.5, which was also typical for other ice cores that experienced large gas loss (Sowers et al., 1989; Bender et al., 1995). The slope of ~0.5 may be the combination of size-dependent fractionation (e.g., 1:1 for $\delta O_2/N_2$ and $\delta Ar/N_2$) and mass-dependent fractionation (1:3 for $\delta O_2/N_2$ and $\delta Ar/N_2$) (Severinghaus et al., 2009). The slope for the DF gas-loss fractionation (~0.2) is significantly smaller than most of the previous values, implying that mass-dependent fractionation

may be more important for the storage condition of the DF core (Fig. 11). Significant mass-dependent fractionation is expected for gas loss through micro-cracks (Bender et al., 1995). Thus, there may have been micro-cracks in the outer ice, although we did not observe visible cracks. The insensitivity of $\delta^{15}N$ to gas loss, consistent with earlier findings, may be explained by viscous flow for $N_2$, or molecular-size-limited diffusion through ice matrix (Severinghaus et al., 2009).

### 4.5 Optimal storage and sampling strategy

We discuss here the recommended practices for the storage and measurement of a newly drilled ice core based on our data. For long-term storage, it is more advantageous to have a larger ice-core cross-section and lower temperature. Based on our

Dome Fuji data (~1 cm from the surface is affected by the gas loss at -50 ˚C after 20 years), a square cross-section of 3×3 cm seems sufficient in a -50 ˚C storage, for sampling a central part (cross-section of 1×1 cm or more) that is unaffected by the

post-coring gas loss. The temperature of -50 ˚C was originally selected for inhibiting the clathrate hydrate dissociation due to the relaxation of ice matrix during long-term ice-core storage (Uchida et al., 1994). To obtain high-quality $\delta O_2/N_2$ and $\delta Ar/N_2$ data, it is recommended to test the real ice-core samples to find sufficient removal thickness. The removal thickness can be determined by examining the pair differences of $\delta O_2/N_2$ with different surface removal thicknesses (e.g., 5 and 8 mm or 3 and 5 mm; Oyabu et al., 2020), which should be within the measurement uncertainty. Five pairs or more for a given combination

of removal thicknesses would be required to make the assessment.

The length of a sample is also an important ice-core-specific factor, especially for reasonably averaging the high scatters in the BCTZ. We speculate that the reasonable sample length in the BCTZ to obtain a clear insolation signal may be more than 50 cm for the Dome Fuji ice core. We note that this length should be different for different ice cores because the thicknesses of the alternating layers of high and low clathrate concentrations should be different at different sites (Lüthi et al., 2010;

Shackleton, 2019). To find a reasonable sample length for a core, it is advisable to continuously measure a ~1-m-long section with a ~2 cm resolution and examine various averaging lengths. The sample length should also be larger than one annual layer thickness to average out the seasonal layering (especially important for the cores with high accumulation rates).

**5 Conclusions**

The variations of $\delta O_2/N_2$ and $\delta Ar/N_2$ within the ice sheet from bubbly ice to clathrate hydrate ice at Dome Fuji are reconstructed without post-coring gas-loss signals. Their variations in the bubbly ice zone, upper BCTZ and below BCTZ show close similarity to the local summer insolation. Our $\delta O_2/N_2$ data from the clathrate hydrate ice zone agree with the previous data after the gas-loss correction (Kawamura et al., 2007), with much less scatters, demonstrating that the original air composition is preserved in the ice core stored at -50 ˚C.


The large scatters in the lower BCTZ may be created by mm-scale vertical inhomogeneity of air composition combined with finite sample length. The insolation signal originally recorded at the bubble close-off remains through the BCTZ, and the insolation signal may be reconstructed by analyzing long ice samples (>50 cm).

Below the BCTZ, the scatters around the orbital-scale variability decrease with depth. The high-resolution analyses of five 50-cm segments show decreasing cm-scale variability of $\delta O_2/N_2$ and $\delta Ar/N_2$ with depth, suggesting diffusive smoothing. A one-dimensional gas diffusion model reproduces the smoothing in this zone with the permeation coefficients of Salamatin et al. (2001).

The slope of the pair differences of $\delta Ar/N_2$ to $\delta O_2/N_2$ is about 1 in the BCTZ, and no correlation is found between those of $\delta^{18}O$ and $\delta O_2/N_2$, suggesting that the $O_2$ and Ar fractionations associated with the bubble-clathrate transition are mostly mass-independent (Fig. 11). On the other hand, the slope for the bubble close-off process is around 0.5, suggesting a combination of mass-independent and mass-dependent fractionation for $O_2$ and Ar. For the BCTZ, the slower molecular diffusion of Ar than $O_2$ may be canceled by the lower dissociation pressure of Ar than $O_2$, which should produce a steeper Ar partial pressure

gradient from the bubbles to the growing clathrate. For the bubble close-off of small molecules, such a cancellation cannot occur, and both the bond-breaking mechanism and the interstitial mechanism may play roles for the molecular diffusion through the thin ice wall of fresh bubbles. The slope is small (0.2) in ice that experienced large gas loss, suggesting that mass-dependent fractionation may be important for the storage condition of the DF core.

The primary application of the $\delta O_2/N_2$ record has been the orbital tuning of the ice-core age scales. In the future, high-precision $\delta O_2/N_2$ and $\delta Ar/N_2$ data of the Dome Fuji core may be obtained with our technique for precise orbital tuning of the ice core. The high-precision data may also provide non-insolation signals on the gases and eventually be useful for reconstructing past atmospheric oxygen and argon concentrations.

More observational and theoretical work is still needed for advancing our understanding of the mechanisms of gas movements in different zones in the ice sheet. For example, the current precisions of ice-core measurements of $\delta^{18}O$ of $O_2$ and $\delta^{40}Ar$ are insufficient for detecting mass-dependent fractionation during bubble close-off and bubble-clathrate transformation processes (note that the mass-dependent fractionation of $\delta^{18}O$ during bubble close-off was evidenced by the WAIS Divide firn-air data, Battle et al., 2011). Theoretical works including molecular dynamics simulations for different gases and ice conditions may

shed light on the different relationships between $\delta O_2/N_2$ and $\delta Ar/N_2$ in different zones. Finally, the International Partnership for Ice Core Sciences regards the retrieval of an ice core containing ice older than 1 million years as highest priority (Fisher et al., 2013; Tsutaki et al., in review). Our constraints on the permeation coefficients of the gases in ice might be useful for predicting the magnitude of diffusive smoothing of air composition in such an ice core.

**Appendix A: Diffusion model**

The time-varying inputs for the model are temperature and thinning function. The temperature is used to calculate dissociation pressure and diffusion constants, and the thinning function is used to determine box size. There is no tunable parameter in the model.

**Parameters for Ar dissociation pressure**

Because there are no published values for $a_{Ar}$ and $b_{Ar}$, we found them by fitting the dissociation pressures of argon hydrate vs. temperature measured by Nagashima et al. (2018) with eq. 4 (Fig. A1). The $a_{Ar}$ and $b_{Ar}$ are 3.63 and 739.5, respectively.

**Molar fraction of *m*-molecule in clathrate hydrate**

The molar fraction of the *m*-molecule in the clathrate hydrate is given by

$$X_m = \frac{U_m}{U_{N_2} + U_{O_2} + U_{Ar} + U_{others}^0}, \tag{A1}$$

where $U_m$ is the concentration of *m*-molecule in total air content. $U_{others}^0$ is the concentration of minor gases, which is assumed to be constant and given by

$$U_{others}^0 = U^0 - U_{N_2}^0 - U_{O_2}^0 - U_{Ar}^0 = \frac{TAC \cdot M_{H_2O}}{V_{STP}} \left(1 - R_{N_2} - R_{O_2} - R_{Ar}\right), \tag{A2}$$

where $U^0$ is concentration of total air content in ice, $U_m^0$ is concentration of *m*-molecule with the atmospheric ratio in the total air content, $TAC$ is total air content, $M_{H_2O}$ is molar mass of ice (H2O), $V_{STP}$ is molar volume of a gas at standard temperature and pressure, and $R_m$ is the atmospheric ratio of *m*-molecule.

**Permeability**

The diffusivity $D_m$, solubility $S_m$, or their product (permeability, $k_m$) for air molecules are taken from the literature (Ikeda-Fukazawa et al., 2001; Salamatin et al., 2001; Ikeda-Fukazawa et al., 2005). Ikeda-Fukazawa et al. (2001) and Salamatin et al. (2001) estimated permeability of N2 and O2, which were constrained by observed δO2/N2 of individual air inclusions in BCTZ. Ikeda-Fukazawa et al. (2005) estimated solubility and diffusivity of N2 and O2 based on molecular dynamics simulations, and the results were consistent with gas-loss fractionation for the Dome Fuji core. The permeability (m² s⁻¹) at $T$ (K) and 1 MPa of Ikeda-Fukazawa et al. (2001) (hereafter IkFk01) is given by

$$k_m = k_m^0 P_m^d \exp\left(-\frac{E_m^k}{RT}\right), \tag{A3}$$

where $k_m^0$ is a constant, $P_m^d$ is dissociation pressure of *m*-molecule, $E_m^k$ is activation energy of permeation for *m*-molecule, $R$ is the gas constant. The permeability (m² s⁻¹) at temperature $T$ (K) and 1MPa of Salamatin et al. (2001) (hereafter Salm01) is given by

$$k_m = k_m^0 \frac{P_m^d}{P_m^{220}} \exp\left[\frac{E_m^k}{R}\left(\frac{1}{220} - \frac{1}{T}\right)\right], \tag{A4}$$

where $P_m^{220}$ is dissociation pressure of *m*-molecule at 220 K. The diffusivity $D_m$ or permeability $k_m$ (m² s⁻¹) at temperature $T$ (K) of Ikeda-Fukazawa et al. (2005) (hereafter IkFk05) is given by

$$D_m = D_m^0 \exp\left(-\frac{E_m^D}{RT}\right). \tag{A5}$$

The solubility at 1MPa of Ikeda-Fukazawa et al. (2005) is given by

$$S_m = S_m^0 \exp\left(-\frac{E_m^S}{RT}\right), \tag{A6}$$

where $S_m^0$ is a constant for *m*-molecule, $E_m^S$ is activation energy of solubility for *m*-molecule. We used those permeation parameters for our model (parameters are summarized in Table A2 and each permeability is shown in Fig. 2 and Table A3).

There are no published values of $k_{Ar}$, thus we estimated it from $k_{N2}$ and $k_{O2}$ in Salamatin et al. (2001) with two formulations by Kobashi et al. (2015). The first one $k_{Ar(I)}$ uses diffusion coefficients of $N_2$, $O_2$ and Ar at 270 K from the molecular dynamic simulations by Ikeda-Fukazawa et al. (2004) ($D_{N_2}^{270}$: 2.1×10⁻¹¹ m² s⁻¹, $D_{O_2}^{270}$: 4.7×10⁻¹¹ m² s⁻¹ and $D_{Ar}^{270}$: 4.0×10⁻¹¹ m² s⁻¹):

$$k_{Ar(I)} = k_{O_2} - \left(\frac{D_{O_2}^{270} - D_{Ar}^{270}}{D_{O_2}^{270} - D_{N_2}^{270}}\right)\left(k_{O_2} - k_{N_2}\right) \qquad \text{(A7)}$$

The second permeability $k_{Ar(II)}$ is based on the observations that δAr/N₂ is often depleted about half of δO₂/N₂ (e.g., Severinghaus et al., 2009). The permeability of Ar(II) is expressed as

$$k_{Ar(II)} = \frac{(k_{N_2} + k_{O_2})}{2}. \qquad \text{(A8)}$$

**Discretization**

The model uses the central differencing scheme. The downward diffusive flux ($f_m$) of $m$-molecule per unit area at the top boundary of $i$-th box is the product of the diffusivity and concentration gradient:

$$f_{m(i)} = D_m \frac{c_{m(i-1)}^h - c_{m(i)}^h}{\Delta z \tau_r}. \qquad \text{(A9)}$$

where $\Delta z$ is initial box height (0.5 mm) and $\tau_r$ is relative thinning function (thinning function divided by the initial value at 1258 m). By substituting eq. 3 into eq. A9, $f_m$ is expressed as

$$f_{m(i)} = \frac{D_m S_m P_m^d}{\Delta z \tau_r}\left(X_{m(i-1)} - X_{m(i)}\right). \qquad \text{(A10)}$$

The net flux of $m$-molecule for $i$-th box is

$$F_{m(i)} = \frac{f_{m(i)} - f_{m(i+1)}}{\Delta z \tau_r}, \qquad \text{(A11)}$$

and the concentration change of $m$-molecule in total air content becomes

$$\Delta U_{m(i)} = F_{m(i)} \Delta t, \qquad \text{(A12)}$$

where $\Delta t$ is time step (~11.6 days).

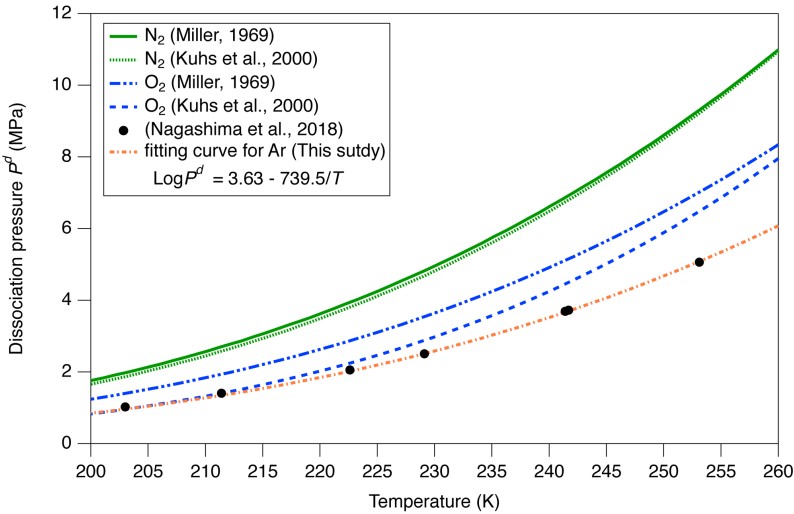

Figure A1: Relationship between temperature and dissociation pressure for $N_2$, $O_2$ (Miller, 1969; Kuhs et al., 2000) and Ar hydrates (Nagashima et al., 2018). The data for Ar hydrate (solid circles) is fitted with an exponential function: $\log P^d = 3.63 - 739.5/T$.

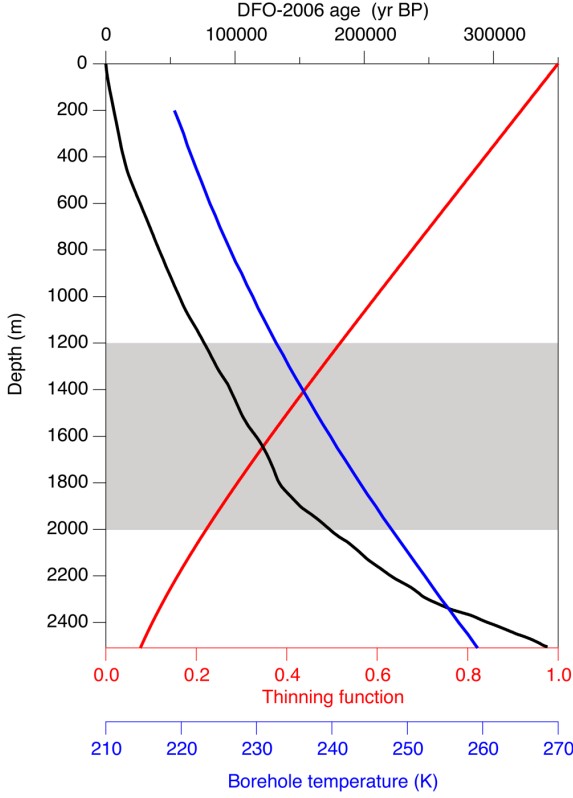

**Figure A2:** Thinning function (red, Nakano et al., 2016), temperature (blue, Buizert et al., 2021) and age (black, Kawamura et al., 2007) in 590 the ice sheet at Dome Fuji used for the diffusion model. Grey shading indicates the depth range of the model run.

**Table A1:** List of symbols

| Symbol | meaning | Unit |
|---|---|---|
| $m$ | molecule species ($m$ = N$_2$, O$_2$ or Ar) | |
| $C_m^h$ | concentration of $m$-molecule | mol mol$_{ice}^{-1}$ |
| $P_m^d$ | dissociation pressure of $m$-molecule | MPa |
| $X_m$ | molar fraction of $m$-molecule in the clathrate hydrate | |
| $a_m$ | constant of dissociation pressure for $m$-molecule | |
| $b_m$ | constant of dissociation pressure for $m$-molecule | K |
| $T$ | temperature | K |
| $U_m$ | concentration of $m$-molecule in total air content | mol mol$_{ice}^{-1}$ |
| $U^0$ | concentration of total air content in ice | mol mol$_{ice}^{-1}$ |
| $U_m^0$ | concentration of $m$-molecule with the atmospheric ratio in the total air content | mol mol$_{ice}^{-1}$ |
| $U_{others}^0$ | concentration of minor gases | mol mol$_{ice}^{-1}$ |
| $TAC$ | total air content | m$^3$ g$^{-1}$ |
| $M_{H_2O}$ | molar mass of ice (H$_2$O) (18.05128) | g mol$^{-1}$ |
| $V_{STP}$ | molar volume of a gas at 273.15K and 100 kPa (0.022711) | m$^3$ mol$^{-1}$ |
| $R_m$ | atmospheric ratio of $m$-molecule | |
| $k_m$ | permeation coefficient of $m$-molecule in ice | m$^2$ s$^{-1}$ |
| $k_m^0$ | constant of permeation for $m$-molecule in ice, | m$^2$ s$^{-1}$ |
| $D_m$ | diffusion coefficient of $m$-molecule in ice | m$^2$ s$^{-1}$ |
| $D_m^0$ | constant of diffusion for $m$-molecule in ice, | m$^2$ s$^{-1}$ |
| $S_m$ | solubility of $m$-molecule in ice | mol mol$_{ice}^{-1}$ MPa$^{-1}$ |
| $S_m^0$ | constant of solution for $m$-molecule in ice | mol mol$_{ice}^{-1}$ MPa$^{-1}$ |
| $E_m^k$ | activation energy of permeation for $m$-molecule in ice | kJ mol$^{-1}$ |
| $E_m^D$ | activation energy of diffusion for $m$-molecule in ice | kJ mol$^{-1}$ |
| $E_m^S$ | activation energy of solubility for $m$-molecule in ice | kJ mol$^{-1}$ |
| $R$ | gas constant (8.314) | kJ mol$^{-1}$ K$^{-1}$ |
| $f_m$ | diffusive flux of $m$-molecule | mol m$^{-2}$ s$^{-1}$ |
| $F_m$ | net flux of $m$-molecule dissolved in ice | mol s$^{-1}$ |
| $\Delta z$ | grid size | m |
| $\tau_r$ | relative thinning function | |
| $\Delta t$ | time step | s |

**Table A2:** Model parameters

|  |  | IkFk01 | Salm01 | IkFk05 |
|---|---|---|---|---|
| constant of dissociation pressure for N$_2$ | $a_{N_2}$ | 3.6905 [a] | 3.6905 [a] | 3.77 [b] |
| constant of dissociation pressure for O$_2$ | $b_{N_2}$ | 688.9 [a] | 688.9 [a] | 4.17 [b] |
| constant of dissociation pressure for N$_2$ (K) | $a_{O_2}$ | 3.679 [a] | 3.67 [a] | 710 [b] |
| constant of dissociation pressure for O$_2$ (K) | $b_{O_2}$ | 717 [a] | 717 [a] | 850 [b] |
| constant of diffusion or permeation for N$_2$ in ice (m$^2$ s$^{-1}$) | $k_{N_2}^0$ | $2.4\times10^{-9}$ [c] | $5.7\times10^{-22}$ [d] | - |
| constant of diffusion or permeation for O$_2$ in ice (m$^2$ s$^{-1}$) | $k_{O_2}^0$ | $2.4\times10^{-8}$ [c] | $1.7\times10^{-21}$ [d] | - |
| constant of solution for N$_2$ in ice (mol mol$_{ice}^{-1}$ MPa$^{-1}$) | $S_{N_2}^0$ | - | - | $4.5\times10^{-7}$ [e] |
| constant of solution for O$_2$ in ice (mol mol$_{ice}^{-1}$ MPa$^{-1}$) | $S_{O_2}^0$ | - | - | $3.7\times10^{-7}$ [e] |
| constant of diffusion for N$_2$ in ice (m$^2$ s$^{-1}$) | $D_{N_2}^0$ | - | - | $2.0\times10^{-10}$ [e] |
| constant of diffusion for O$_2$ in ice (m$^2$ s$^{-1}$) | $D_{O_2}^0$ | - | - | $3.5\times10^{-9}$ [e] |
| activation energy of permeation for N$_2$ in ice (kJ mol$^{-1}$) | $E_{N_2}^k$ | 55.57 [c] | 50 [d] | - |
| activation energy of permeation for O$_2$ in ice (kJ mol$^{-1}$) | $E_{O_2}^k$ | 55.57 [c] | 50 [d] | - |
| activation energy of solution for N$_2$ in ice (kJ mol$^{-1}$) | $E_{N_2}^S$ | - | - | 9.2 [e] |
| activation energy of solution for O$_2$ in ice (kJ mol$^{-1}$) | $E_{O_2}^S$ | - | - | 7.9 [e] |
| activation energy of diffusion for N$_2$ in ice (kJ mol$^{-1}$) | $E_{N_2}^D$ | - | - | 5.1 [e] |
| activation energy of diffusion for O$_2$ in ice (kJ mol$^{-1}$) | $E_{O_2}^D$ | - | - | 9.7 [e] |

[a] Miller (1969), [b] Kuhs et al. (2000), [c] Ikeda-Fukazawa et al. (2001), [d] Salamatin et al. (2001), [e] Ikeda-Fukazawa et al. (2005).

**Table A3:** Examples of dissociation pressure, diffusivity, solubility and permeability (at 240 K)

|  |  | IkFk01 | Salm01 | IkFk05 | Ar (I) | Ar (II) |
|---|---|---|---|---|---|---|
| Dissociation pressure of N$_2$ (MPa) | $P_{N_2}^d$ | 6.48[a] | 6.61[b] | 6.48 [a] |  |  |
| Dissociation pressure of O$_2$ (MPa) | $P_{O_2}^d$ | 4.25 [a] | 4.92 [b] | 4.25 [a] |  |  |
| Dissociation pressure of Ar (MPa) | $P_{Ar}^d$ |  |  |  | 3.54 | 3.54 |
| Diffusivity of N$_2$ (m$^2$ s$^{-1}$) | $D_{N_2}$ |  |  | $1.55\times10^{-11}$ |  |  |
| Diffusivity of O$_2$ (m$^2$ s$^{-1}$) | $D_{O_2}$ |  |  | $2.71\times10^{-11}$ |  |  |
| Solubility of N$_2$ (mol mol$_{ice}^{-1}$ MPa$^{-1}$) | $S_{N_2}$ |  |  | $4.48\times10^{-9}$ |  |  |
| Solubility of O$_2$ (mol mol$_{ice}^{-1}$ MPa$^{-1}$) | $S_{O_2}$ |  |  | $7.06\times10^{-9}$ |  |  |
| Permeability of N$_2$ (m$^2$ s$^{-1}$ MPa$^{-1}$) | $k_{N_2}$ | $1.27\times10^{-20}$ | $1.02\times10^{-20}$ | $6.95\times10^{-20}$ |  |  |
| Permeability of O$_2$ (m$^2$ s$^{-1}$ MPa$^{-1}$) | $k_{O_2}$ | $9.49\times10^{-20}$ | $3.12\times10^{-20}$ | $1.91\times10^{-19}$ |  |  |
| Permeability of Ar (m$^2$ s$^{-1}$ MPa$^{-1}$) | $k_{Ar}$ |  |  |  | $2.55\times10^{-20}$ | $2.07\times10^{-20}$ |

[a] Miller (1969), [b] Kuhs et al. (2000).

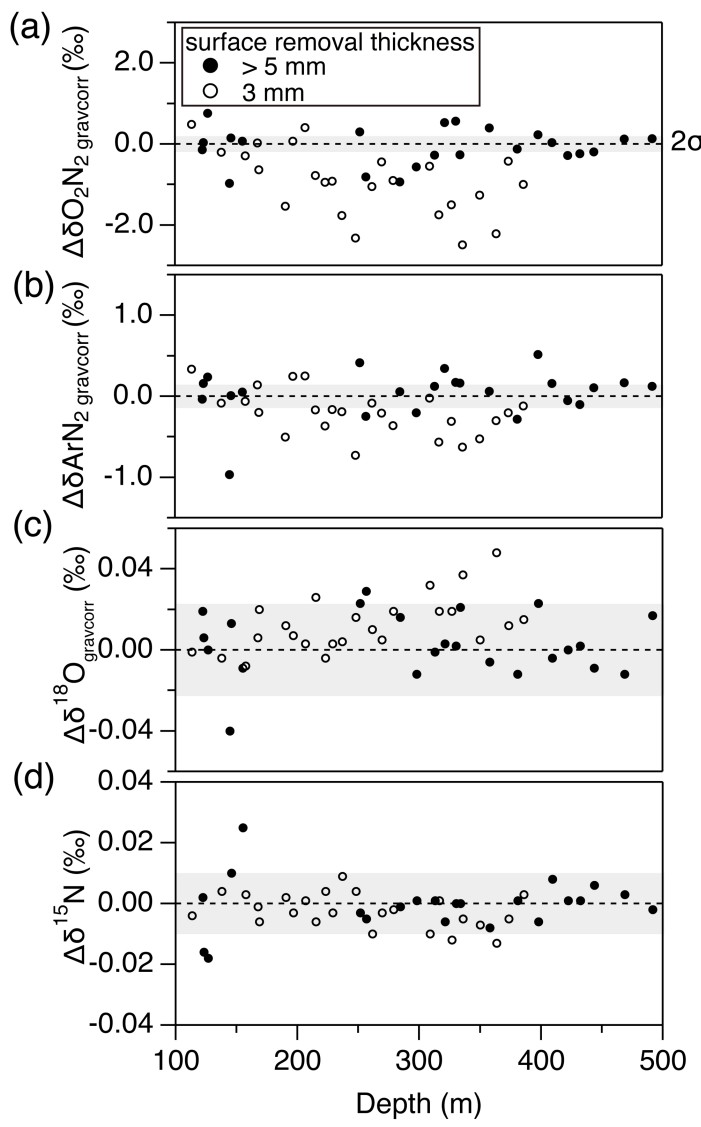


**Figure B1:** Pair difference ("a2" piece minus "a1" piece, Fig. 1) in the bubbly ice for (a) $\delta O_2/N_2$, (b) $\delta$ Ar/$N_2$ (c) $\delta^{18}O$ and (d) $\delta^{15}N$. Solid circles are the data from the samples after removing the outer ice by > 5-mm thickness, and open circles are the data with ~3-mm removal. Grey shadings are 2 pooled standard deviations of the pair differences for clathrate ice, shown as uncertainty estimates.


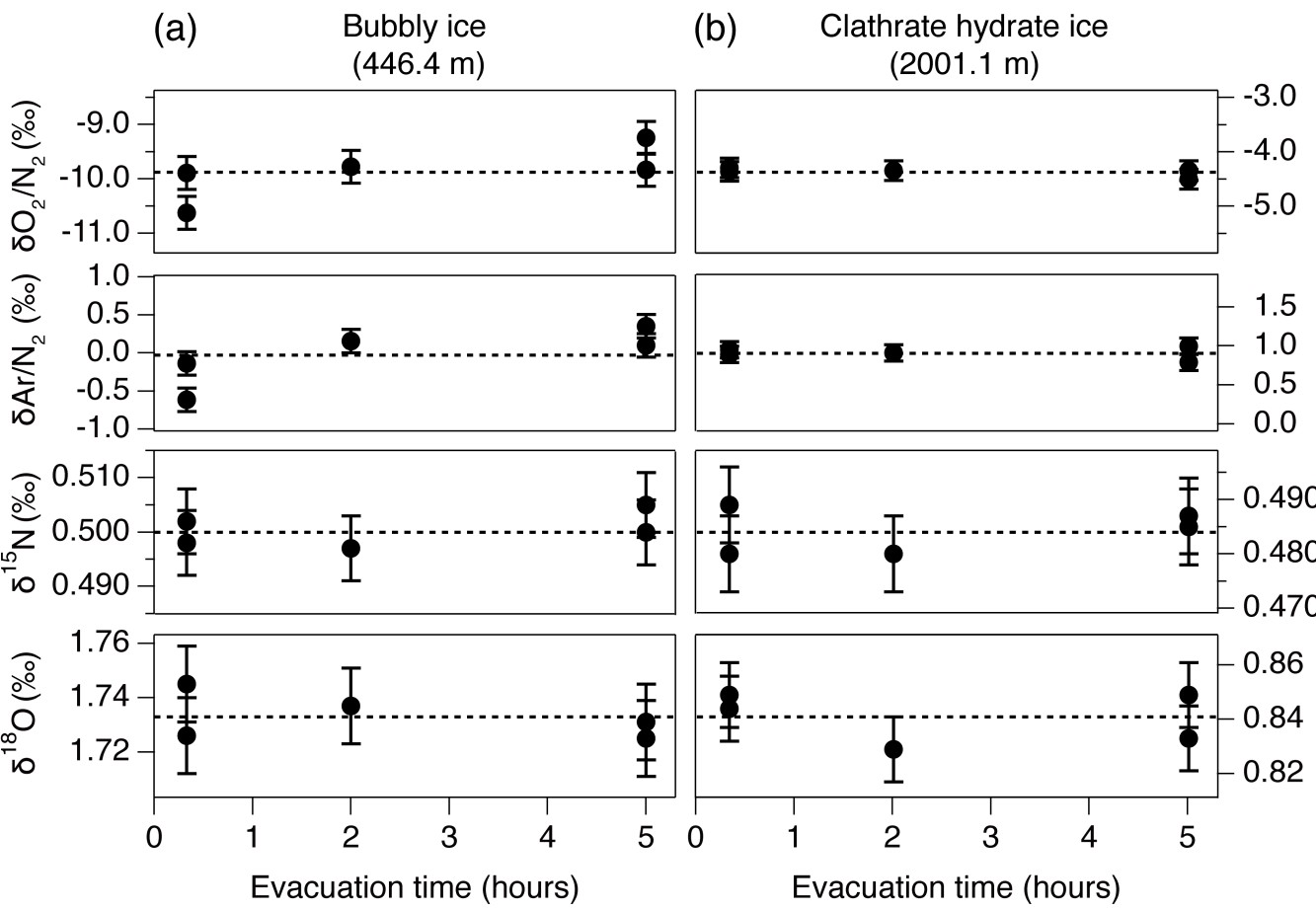

**Figure B2:** Results of the evacuation experiment for $\delta O_2/N_2$, $\delta Ar/N_2$, $\delta^{15}N$ and $\delta^{18}O$ in (a) bubbly ice (446.4 m) and (b) clathrate hydrate ice (2001.1 m). Dashed lines are the average of all 5 values.

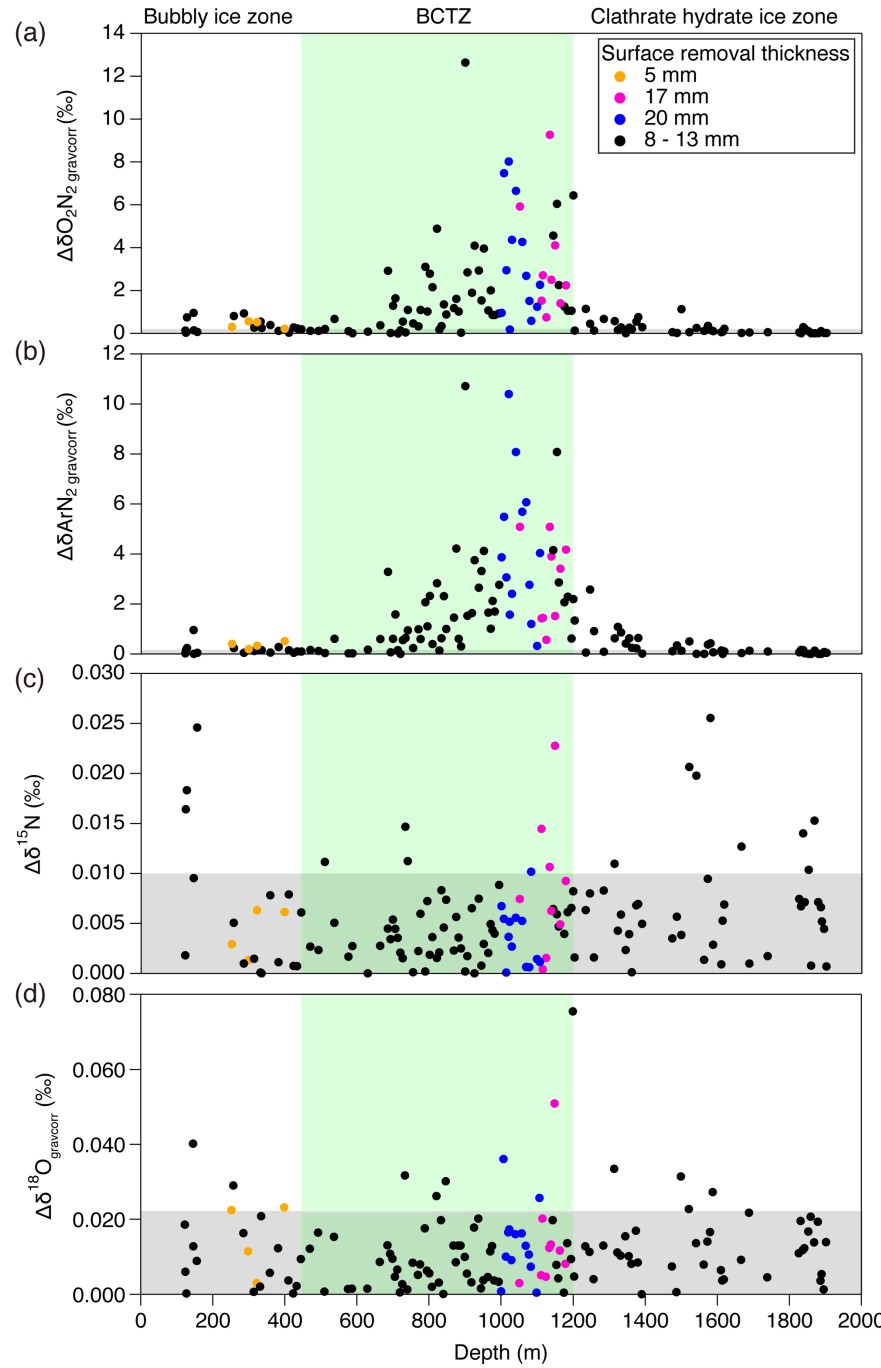


**Figure B3:** Absolute pair differences of (a) $\delta O_2/N_{2gravcorr}$, (b) $\delta Ar/N_{2gravcorr}$, (c) $\delta^{15}N$ and (d) $\delta^{18}O_{gravcorr}$. Grey shadings indicate the 2 pooled standard deviations for clathrate hydrate ice.

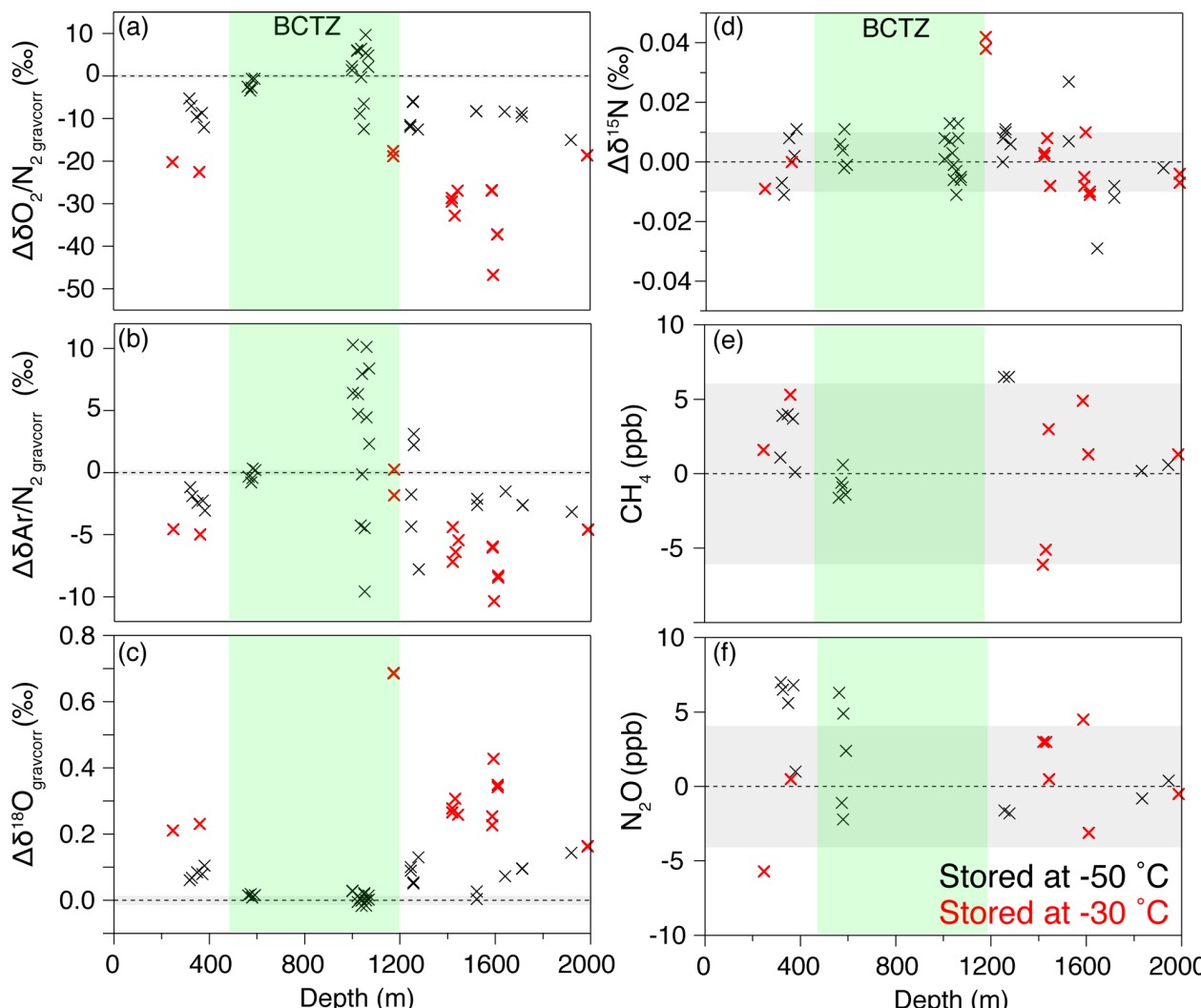

**Figure B4:** Differences of the values in the outer and inner pieces (outer minus inner) for (a) $\delta O_2/N_{2gravcorr}$, (b) $\delta Ar/N_{2gravcorr}$, (c) $\delta^{18}O_{gravcorr}$, (d) $\delta^{15}N$, (e) $CH_4$ concentration, and (f) $N_2O$ concentration. Grey shadings indicate $2\sigma$ pooled standard deviations of pair differences (from inner ice only) in the clathrate hydrate ice. Black and red markers are the data using the outer samples stored at -50 ˚C and -30 ˚C, respectively.

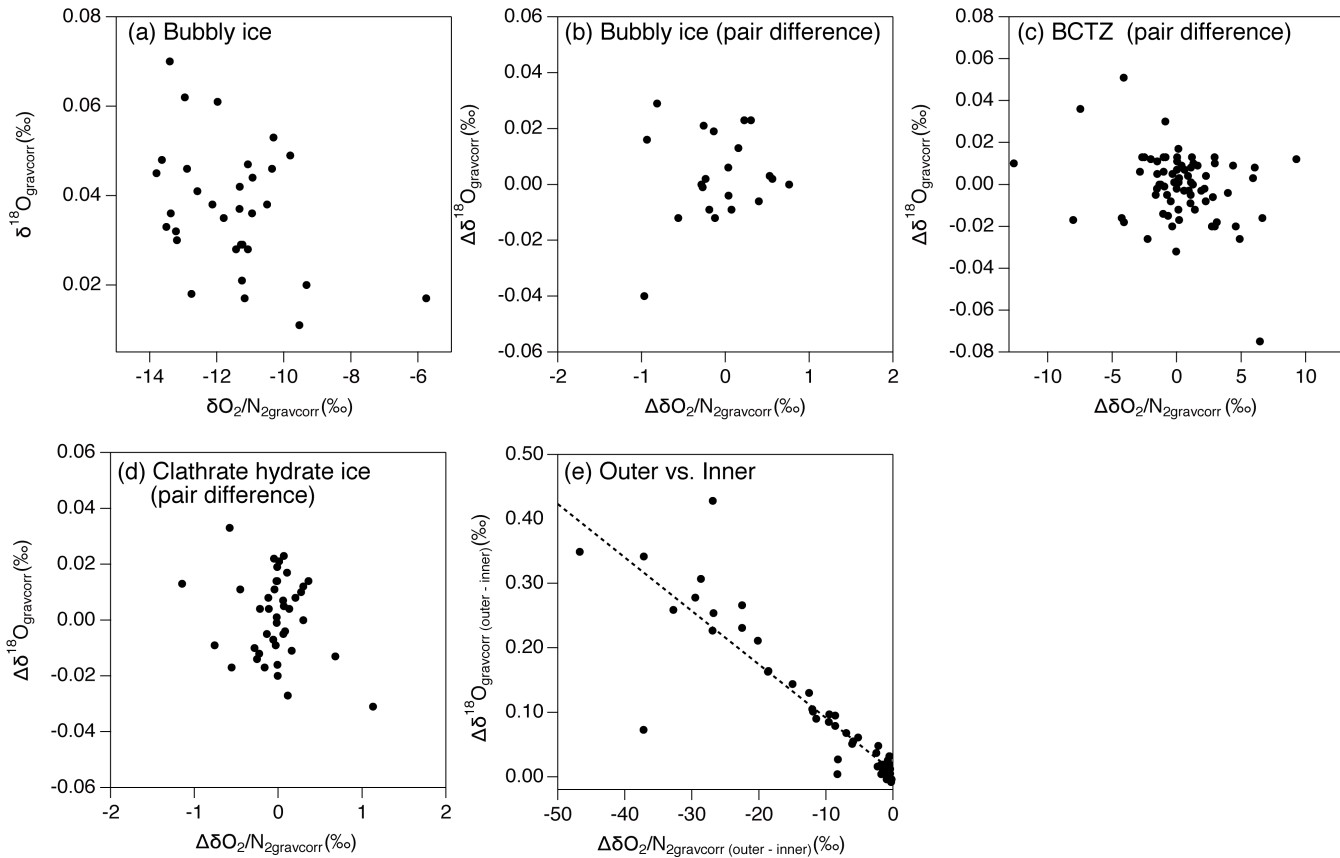

**Figure B5:** Relationship between $\delta O_2/N_{2gravcorr}$ and $\delta^{18}O_{gravcorr}$ for (a) the last 2000 years for bubbly ice, (b) pair differences in bubbly ice zone, (c) pair differences in BCTZ, (d) pair differences in clathrate hydrate ice zone, and (e) differences between the outer ice and inner ice (bubbly ice and clathrate hydrate ice). Dashed line represents linear fit to the data ($\Delta\delta^{18}O_{gravcorr} = -0.0083\Delta\delta O_2/N_{2gravcorr} + 0.0018$).


## Appendix C: Supplementary tables

**Table C1:** Pooled standard deviations for $\delta^{15}N$, $\delta O_2/N_{2\text{gravcorr}}$ and $\delta Ar/N_{2\text{gravcorr}}$ (‰).

|  | $\delta^{15}N$ | $\delta O_2/N_{2\text{gravcorr}}$ | $\delta Ar/N_{2\text{gravcorr}}$ | n |
|---|---|---|---|---|
| 112 – 450 m (bubbly ice) | 0.006 | 0.305 | 0.154 | 20 |
| 450 – 800 m (upper BCTZ) | 0.004 | 0.862 | 0.776 | 24 |
| 800 – 1200 m (lower BCTZ) | 0.005 | 2.662 | 2.555 | 46 |
| 1200 – 1480 m (clathrate hydrate ice) | 0.004 | 0.599 | 0.565 | 13 |
| 1480 – 1980 m (clathrate hydrate ice) | 0.007 | 0.184 | 0.112 | 26 |

**Table C2:** Slopes for $\delta Ar/N_{2\text{gravcorr}}$ vs. $\delta O_2/N_{2\text{gravcorr}}$ and for $\Delta\delta Ar/N_{2\text{gravcorr}}$ vs. $\Delta\delta O_2/N_{2\text{gravcorr}}$.

|  | Site name | Zone | Remarks | Slope | 1 σ | Reference for the original data |
|---|---|---|---|---|---|---|
| | Dome Fuji | All (112 – 1980 m) | well preserved | 0.51 | 0.01 | This study |
| | Dome Fuji | Bubbly ice (112 – 450m) | well preserved | 0.50 | 0.01 | This study |
| | Dome Fuji | Upper BCTZ (450 – 800 m) | well preserved | 0.45 | 0.01 | This study |
| | Dome Fuji | Lower BCTZ (800 – 1200 m) | well preserved | 0.61 | 0.01 | This study |
| | Dome Fuji | Clathrate hydrate (1200 – 1980 m) | well preserved | 0.42 | 0.02 | This study |
| | Dome Fuji | Outer ice | significantly affected by gas loss | 0.25 | 0.01 | This study |
| | 9 sites[a] | Bubbly ice | | 0.46 | 0.03 | Sowers et al. (1989) |
| | Byrd | All (145 – 2100 m) | extremely negative below 1800 m | 0.42 | 0.01 | Bender et al. (1995) |
| | Vostok-3G | All (140 – 2058 m) | thermally drilled | 0.70 | 0.01 | Bender et al. (1995) |
| $\delta Ar/N_2/\delta O_2/N_2$ | Dome C | Bubbly ice (180 – 625 m) | poor quality | 0.26 | 0.01 | Bender et al. (1995) |
| | GISP2 | Bubbly ice (73 – 213 m) | | 0.54 | 0.03 | Bender et al. (1995) |
| | Siple Dome | Bubbly ice (75 – 973 m) | highly fractured | 0.58 | 0.01 | Severinghaus et al. (2009) |
| | WAIS Divide | All (80 – 2625 m) | slightly affected by gas loss | 0.93 | 0.01 | Seltzer et al. (2017) |
| | WAIS Divide | Bubbly ice (80 – 2625 m) | slightly affected by gas loss | 0.64 | 0.01 | Seltzer et al. (2017) |
| | WAIS Divide | BCTZ (650 – 1600 m) | slightly affected by gas loss | 1.00 | 0.01 | Seltzer et al. (2017) |
| | WAIS Divide | Clathrate hydrate (1600 – 2625 m) | slightly affected by gas loss | 0.97 | 0.01 | Seltzer et al. (2017) |
| | SPICE | Bubbly ice (125 – 700 m) | slightly affected by gas loss | 0.60 | 0.05 | Severinghaus (2019) |
| | SPICE | BCTZ (700 – 1140 m) | slightly affected by gas loss | 1.47 | 0.07 | Severinghaus (2019) |
| | SPICE | Clathrate hydrate (1140 – m) | slightly affected by gas loss | 0.54 | 0.01 | Severinghaus (2019) |
| | NEEM | Bubbly ice (112 – 460 m) | slightly affected by gas loss | 0.45 | 0.01 | Oyabu et al. (2020) |

| | | | | | | |
|---|---|---|---|---|---|---|
| | Dome Fuji | Bubbly ice (112 – 450m) | well preserved | 0.53 | 0.08 | This study |
| | Dome Fuji | Upper BCTZ (450 – 800 m) | well preserved | 0.92 | 0.03 | This study |
| | Dome Fuji | Lower BCTZ (800 – 1200 m) | well preserved | 1.01 | 0.03 | This study |
| | Dome Fuji | Clathrate hydrate (1200 – 1980 m) | well preserved | not detectable | | This study |
| | Dome Fuji | Outer piece vs inner piece | | 0.22 | 0.01 | This study |
| | Byrd | All (145 – 2100 m) | extremely negative below 1800 m | 0.47 | 0.01 | Bender et al. (1995) |
| | Vostok-3G | All (140 – 2058 m) | thermally drilled | 0.54 | 0.03 | Bender et al. (1995) |
| $\Delta\delta Ar/N_2/\Delta\delta O_2/N_2$ (pair difference) | Dome C | Bubbly ice (180 – 625 m) | poor quality | 0.24 | 0.03 | Bender et al. (1995) |
| | Siple Dome | Bubbly ice (75 – 973 m) | highly fractured | 0.67 | 0.01 | Severinghaus et al. (2009) |
| | WAIS Divide | Bubbly ice (80 – 650 m) | slightly affected by gas loss | 0.72 | 0.01 | Seltzer et al. (2017) |
| | WAIS Divide | BCTZ (650 – 1600 m) | slightly affected by gas loss | 1.19 | 0.01 | Seltzer et al. (2017) |
| | WAIS Divide | Clathrate hydrate (1600 – 3396 m) | slightly affected by gas loss | 2.10 | 0.01 | Seltzer et al. (2017) |
| | SPICE | Bubbly ice ( - 700 m) | slightly affected by gas loss | 0.84 | 0.06 | Severinghaus (2019) |
| | SPICE | BCTZ (700 – 1140 m) | slightly affected by gas loss | 0.88 | 0.06 | Severinghaus (2019) |
| | SPICE | Clathrate hydrate (1140 – m) | slightly affected by gas loss | 1.60 | 0.08 | Severinghaus (2019) |
| | NEEM | Bubbly ice (112 – 460 m) | slightly affected by gas loss | 0.62 | 0.01 | Oyabu et al. (2020) |

[a] Byrd, Camp Century, Crete, Dominion Range, Dye 3, D10, D57 and Dome C ice cores.

## Data availability

All data presented in this study is available at the NIPR ADS data repository (https://ads.nipr.ac.jp/dataset/A20210430-001; https://doi.org/10.17592/001.2021043001, Oyabu et al., 2021) and will be available on the NOAA paleoclimate database.

## Author contribution

IO: Conceptualization, Methodology, Validation, Formal analysis, Investigation, Data Curation, Writing - Original Draft, Visualization, Funding acquisition. KeK: Conceptualization, Methodology, Validation, Investigation, Writing - Original Draft, Supervision, Project administration, Funding acquisition. TU: Writing - Review & Editing. SF: Writing - Review & Editing. KyK: Methodology, Validation, Investigation. MH: Validation, Investigation, SA: Methodology, Validation, Investigation. SM: Methodology, Validation, Investigation. TN: Methodology, Validation, Investigation. JS: Investigation, Writing - Review & Editing, JM: Formal analysis, Investigation, Writing - Review & Editing.

## Competing interests

The authors declare that they have no conflict of interest.

## Acknowledgments

We are grateful to Hiroshi Ohno of Kitami Institute of Technology for providing air inclusion data and comments, Tomoko Fukazawa of Meiji University for providing the Raman $O_2/N_2$ data, Kumiko Goto-Azuma of National Institute of Polar Research for the management of ice-core chemistry analyses, Hirohisa Matsushima, Satoko Nakanishi, Mariko Hayakawa, Akito Tanaka, Satomi Oda, Maki Nakatani, Yasuko Fukagawa, Miho Arai and Ryo Inoue for ice core measurements. We acknowledge the Dome Fuji ice core projects, the Japanese Antarctic Research Expedition, and Ice Core Consortium. This study was supported by Japan Society for the Promotion of Science (JSPS) and Ministry of Education, Culture, Sports, Science and Technology-Japan (MEXT) KAKENHI Grant Numbers 20H04327, 17K12816 and 17J00769 to IO, 20H00639, 17H06320, 15KK0027 and 26241011 to KeK, 18H05294 to SF, and by NIPR Research Projects (Senshin Project and KP305). We are grateful to the two anonymous reviewers for their constructive comments.

650

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

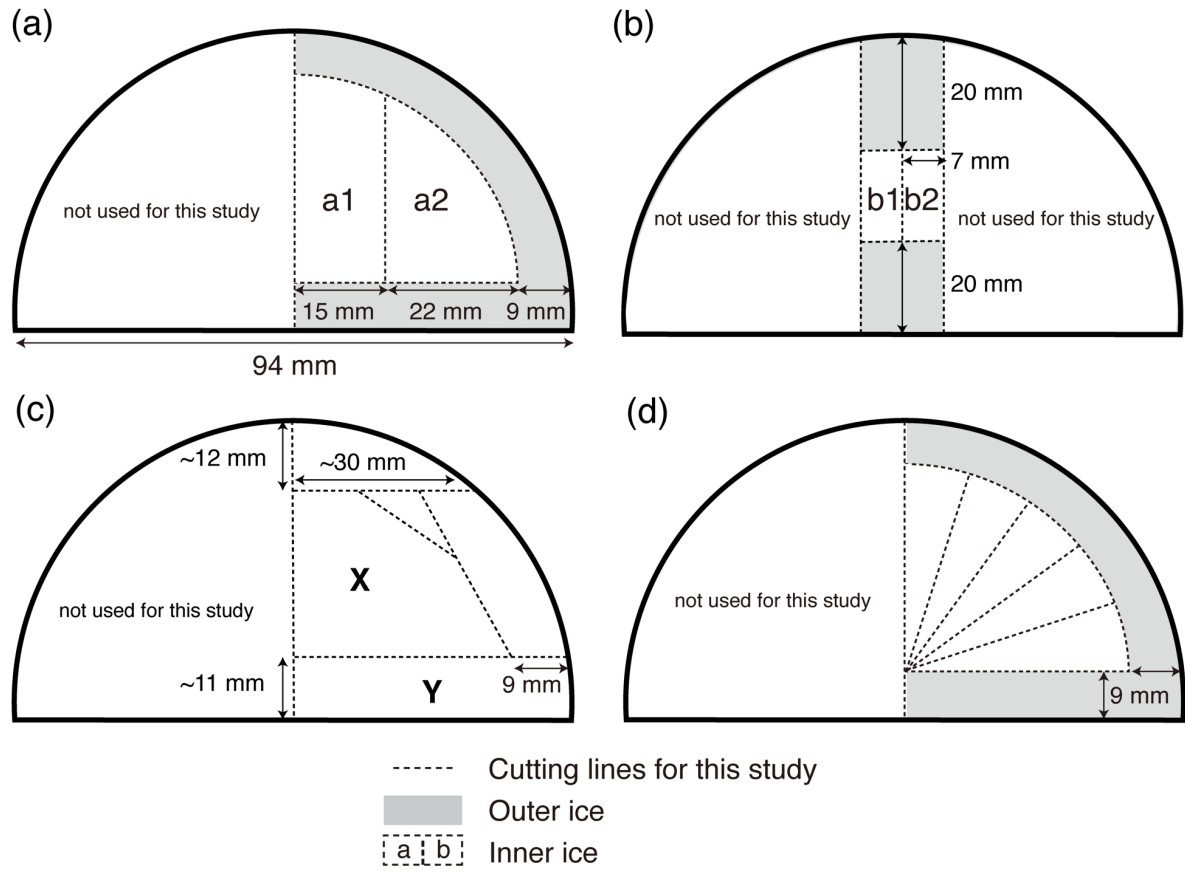

**Figure 1:** Typical cross-sectional cutting plan. The outer black line in each panel is the original surface that was exposed to the atmosphere for ~20 years at -50 °C. Dotted lines are the cutting lines for this study. (a) Duplicate measurements and outer-inner comparisons. The sample length is 11 cm. For single (non-duplicate) measurements, only a2 is cut out. (b) Same as (a) but for 13 depths in 1000 – 1100 m with the sample length of 20 cm. (c) High-resolution analysis. Sections X and Y were used for the gas and ion measurements with the lengths of 2.5 and 1.25 cm, respectively. for X and Y, respectively. (d) Test of gas-loss under vacuum with the length of 11 cm.

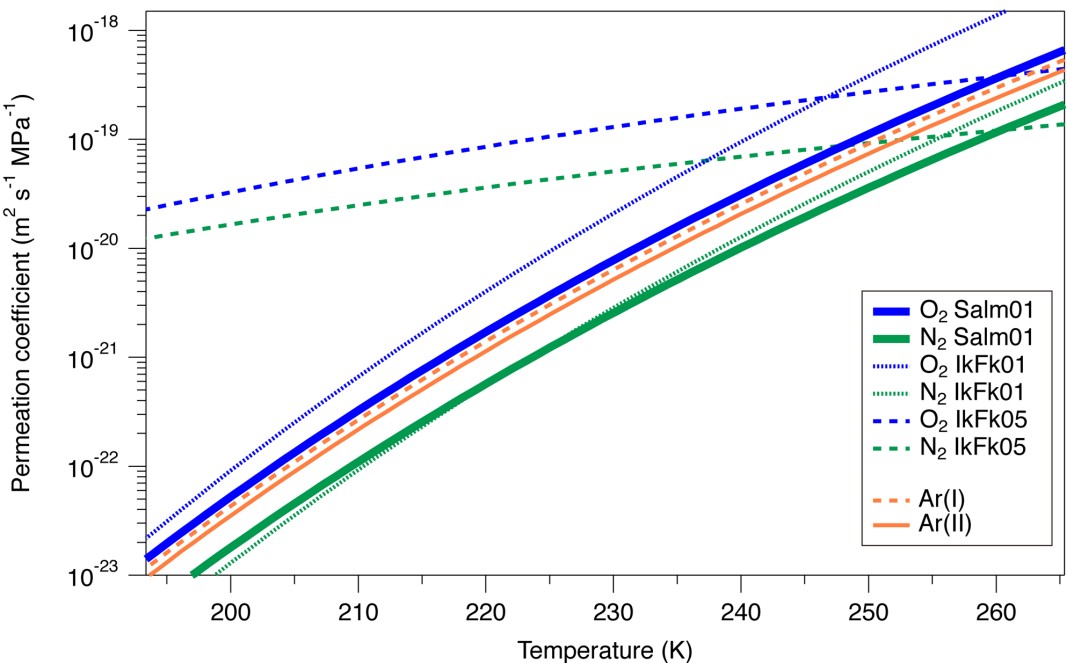

**Figure 2:** Permeation coefficients of $O_2$, $N_2$, Ar and air, estimated by different authors. See text for details.

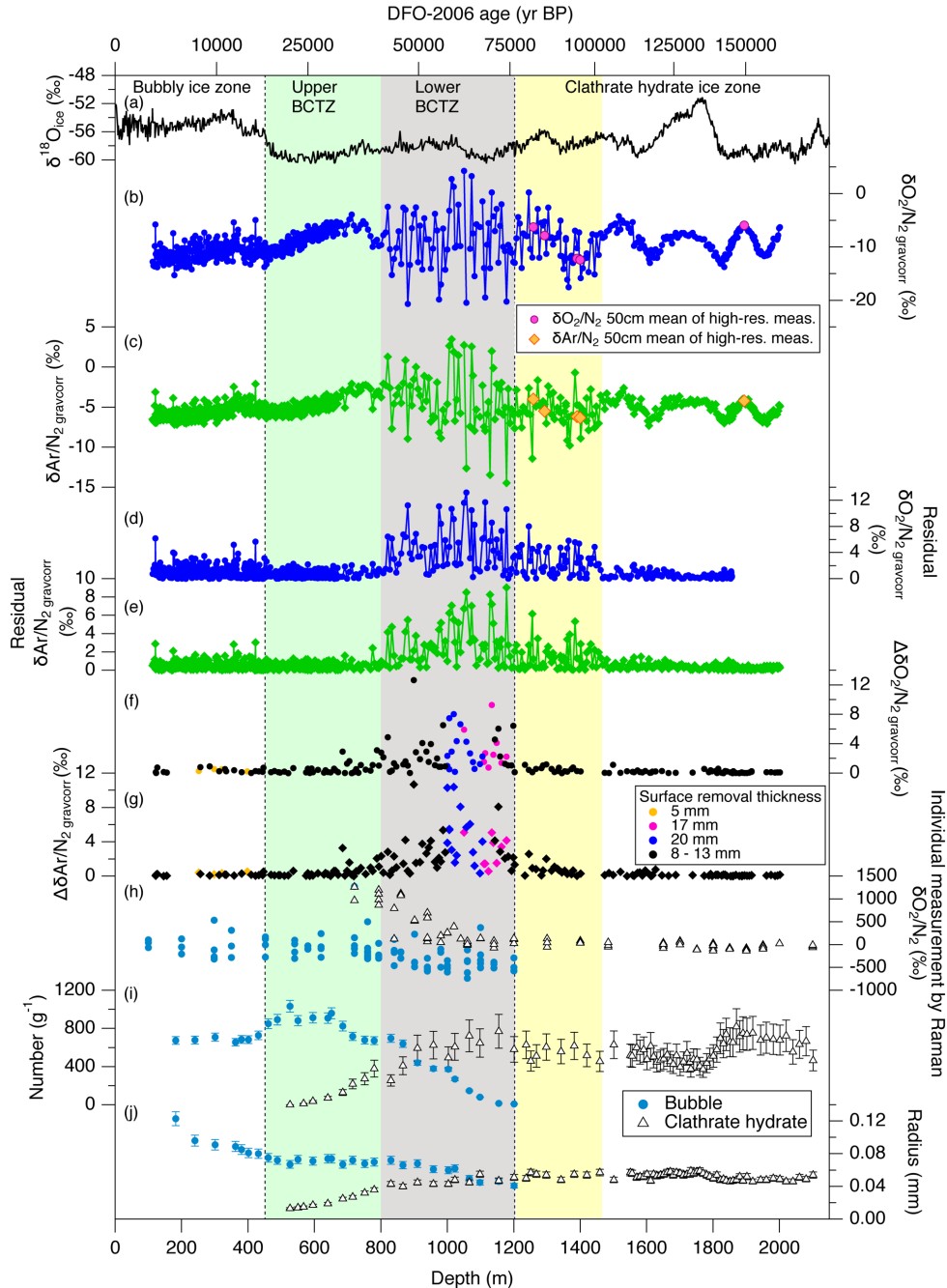

**Figure 3:** $\delta O_2/N_{2gravcorr}$ and $\delta Ar/N_{2gravcorr}$ records from the Dome Fuji ice core. Published data of water isotopes and air inclusions are also shown. (a) Stable water isotope record (Uemura et al., 2018), (b), (c) $\delta O_2/N_{2gravcorr}$ and $\delta Ar/N_{2gravcorr}$ after gravitational corrections, (d), (e) absolute residuals of $\delta O_2/N_{2gravcorr}$ and $\delta Ar/N_{2gravcorr}$ from their low-pass filtered curves, (f), (g) absolute pair differences of $\delta O_2/N_{2gravcorr}$ and $\delta Ar/N_{2gravcorr}$, (h) $\delta O_2/N_2$ in bubbles and clathrate hydrates observed by Raman spectroscopy (Ikeda-Fukazawa et al., 2001), and (i), (j) number concentrations and radius of bubbles and clathrate hydrates (Ohno et al., 2004).

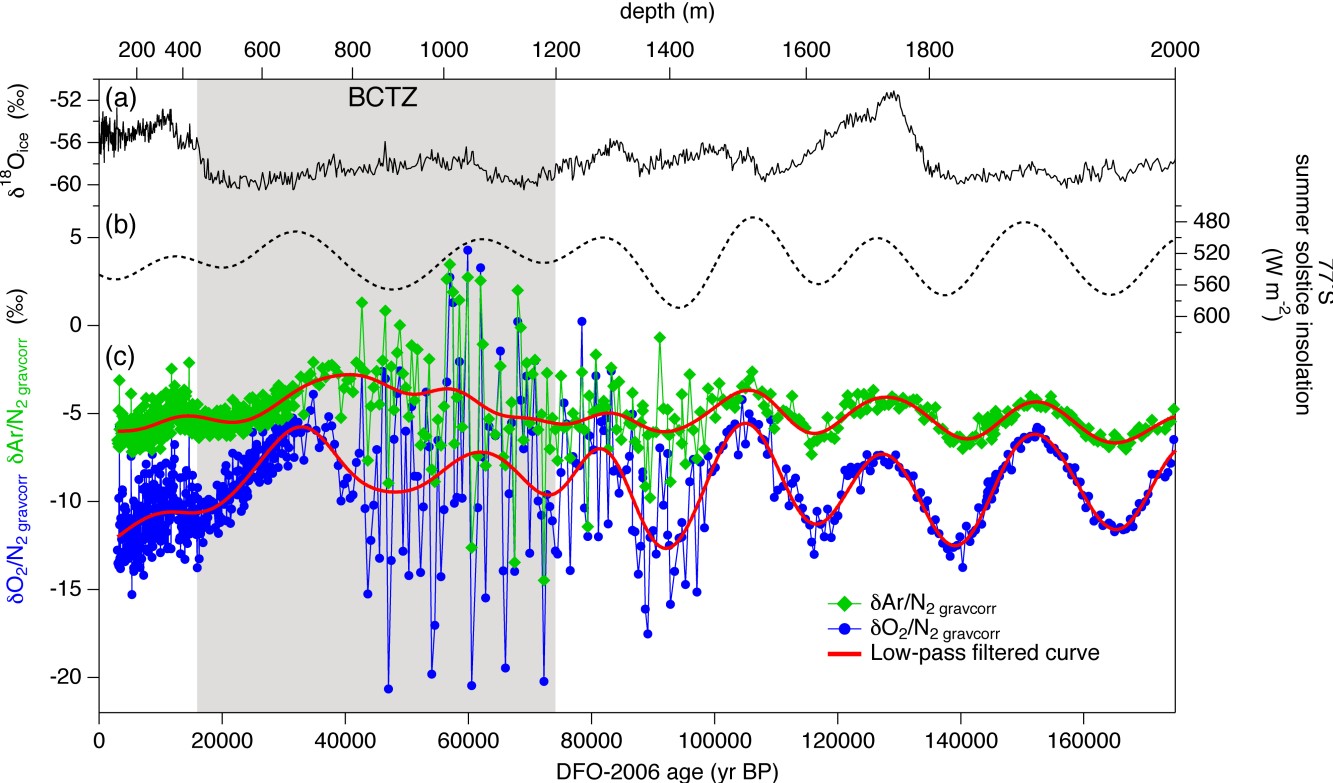


**Figure 4:** $\delta O_2/N_{2gravcorr}$ and $\delta Ar/N_{2gravcorr}$ with published records plotted against the age of ice. (a) Stable water isotope record (Uemura et al., 2018), (b) summer solstice insolation at Dome Fuji (Laskar et al., 2004), and (c) $\delta O_2/N_{2gravcorr}$ and $\delta Ar/N_{2gravcorr}$ with low-pass filtered curves (cut-off at 16.7 kyr, the filter was designed by Kawamura et al., 2007) (red lines).


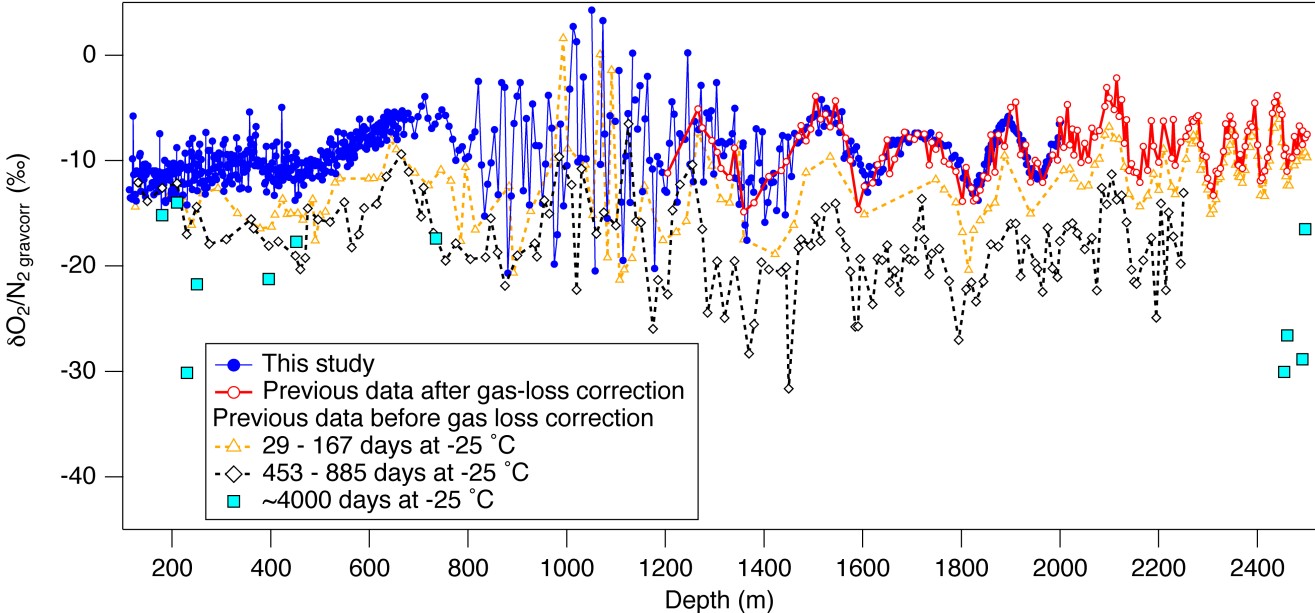

**Figure 5:** Comparison of the new and previous $\delta O_2/N_{2gravcorr}$ data from the Dome Fuji ice core. Solid circles (blue) are the new data (this study), and open circles (red) are the previous data corrected for gas loss during storage at -25 ˚C (Kawamura et al., 2007). Open triangles (orange), open diamonds (black), and solid square (light blue) are the uncorrected previous data for the storage period of 29 – 167 days, 453 – 885 days (Kawamura, 2001) and ~4000 days (11 years) (this study), respectively.

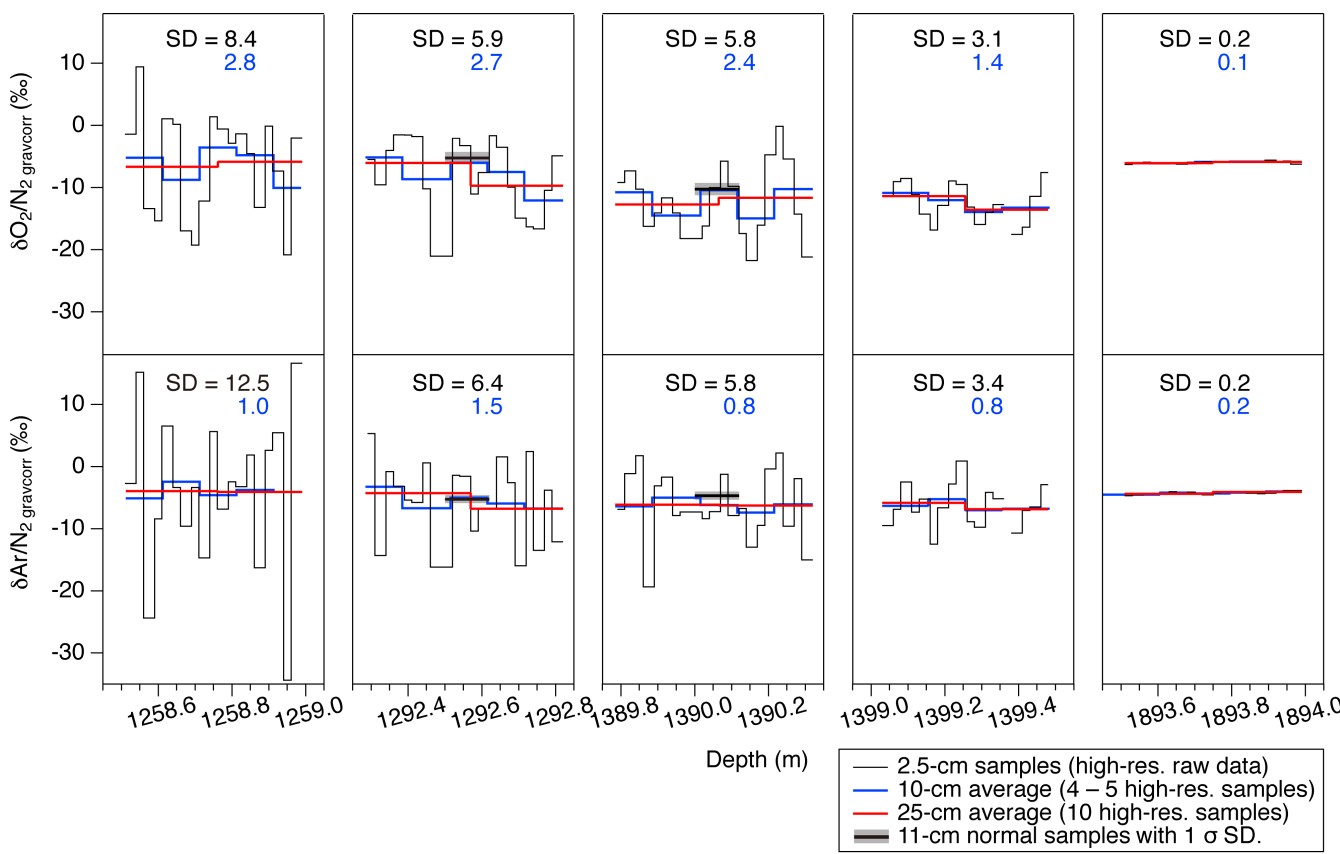

**Figure 6:** High-resolution data of $\delta O_2/N_{2gravcorr}$ and $\delta Ar/N_{2gravcorr}$ in 5 depth intervals. Black, blue and red lines are the raw data (2.5-cm resolution), 10-cm averages and 25-cm averages, respectively. Standard deviations (SD) of the raw 2.5-cm data (black) and 10-cm averages (blue) are shown in each panel. Thick horizontal black bars with gray shadings in some panels are the independent data from the 11-cm samples with 1σ error (measured by the standard protocol).

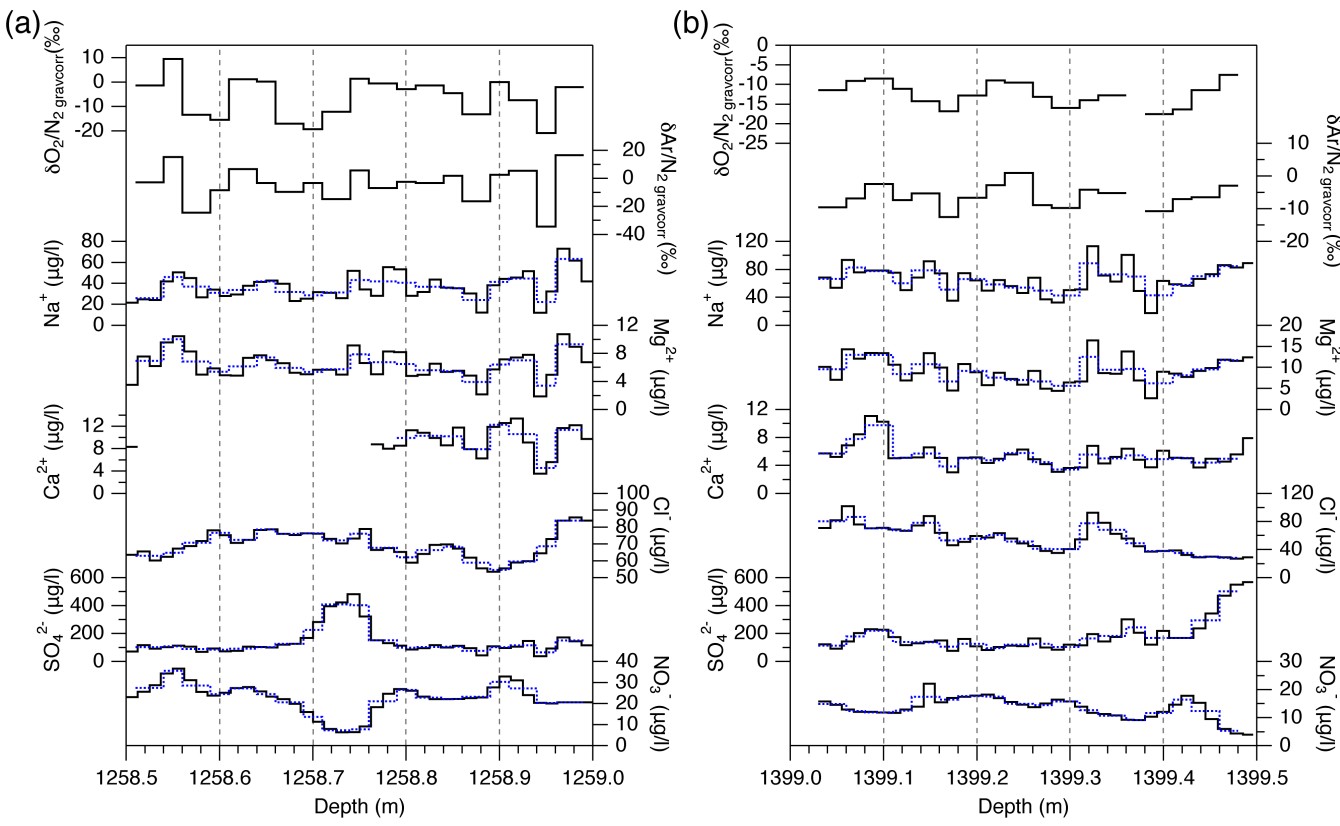


**Figure 7:** High-resolution data of δO₂/N₂gravcorr and δAr/N₂gravcorr (2.5-cm resolution) and ion concentrations (1.25-cm resolution). Blue dotted lines are the interpolated ion data to match with the resolution of the gas data. The Ca²⁺ data for 1258.51 – 1258.75 m are excluded (high contamination due to sample handling error).

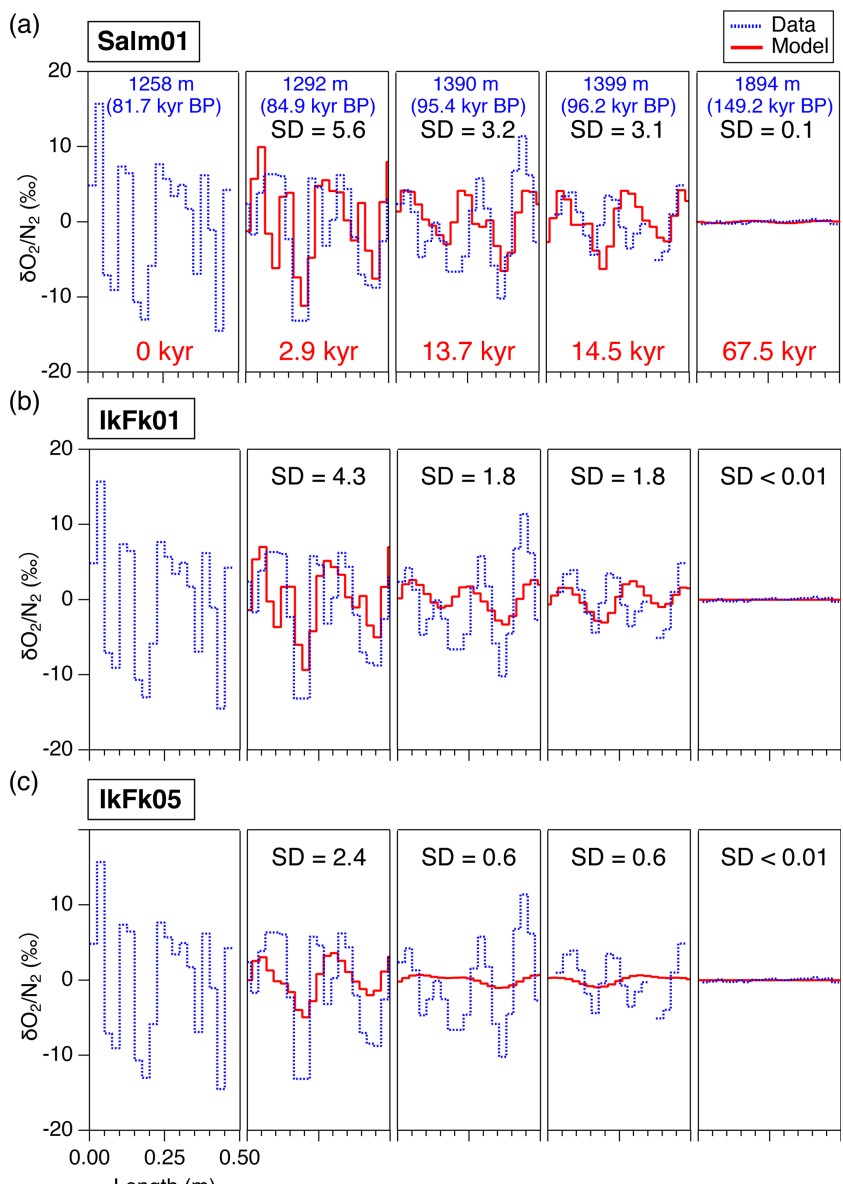

**Figure 8:** Comparison of the diffusion model outputs for $\delta O_2/N_2$ with different permeation parameters (red lines) and high-resolution data (blue dotted lines), at 1292 m, 1390 m, 1399 m and 1894 m. The data at 1258 m were given to the diffusion model as the initial condition. Solid lines (red) are the model outputs at the elapsed time of 2.9, 13.7, 14.5, 67.5 kyr, resampled at 2.5-cm intervals, to compare with the corresponding high-resolution data. Standard deviations (SD) of the model results are shown in each panel.

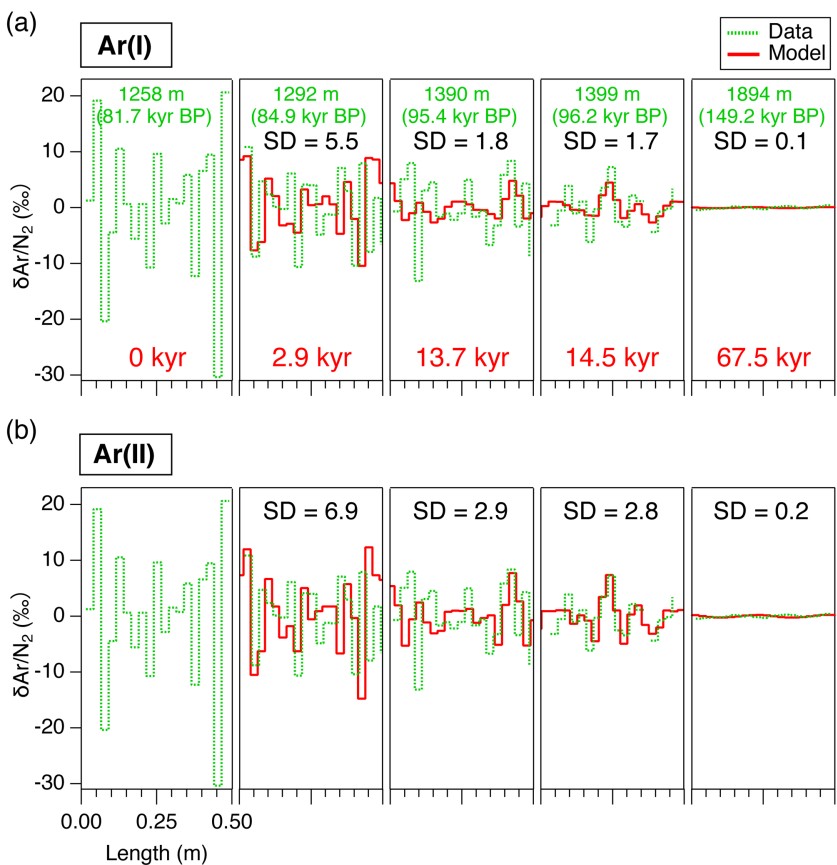


**Figure 9:** Same as Figure 8 but for δAr/N₂.

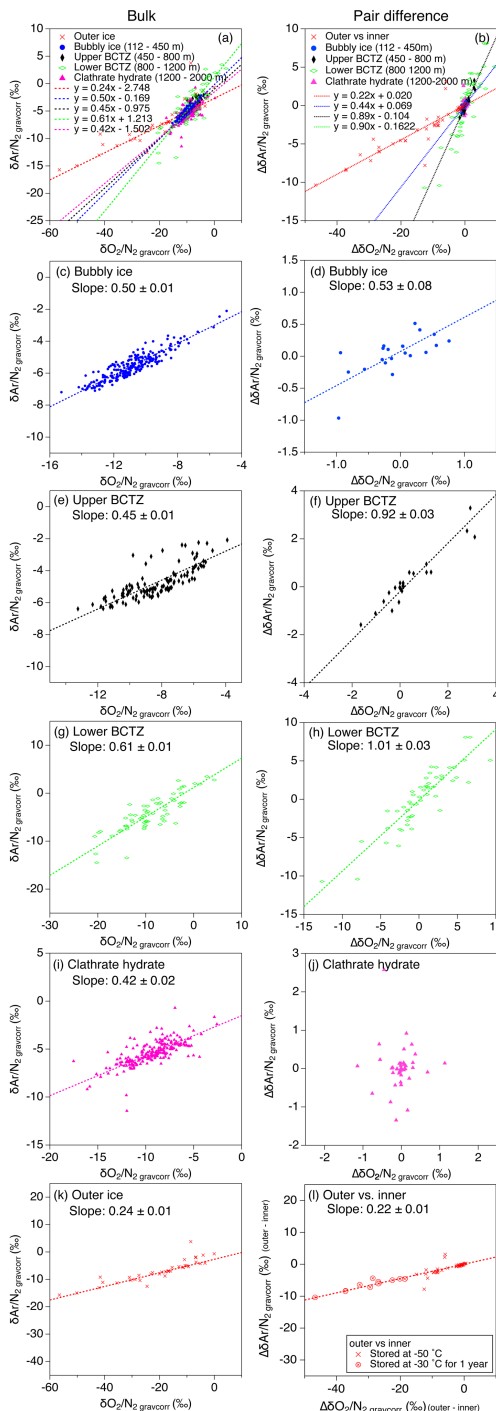

**Figure 10:** Scatter plot of $\delta O_2/N_{2gravcorr}$ vs. $\delta Ar/N_{2gravcorr}$ ((a), (c), (e), (g), (i), (k)), and that of pair differences of $\delta O_2/N_{2gravcorr}$ and $\delta Ar/N_{2gravcorr}$ ((b), (d), (f), (h), (j), (l)). (a) and (b) all data, (c) and (d) bubbly ice zone (112 – 450 m), (e) and (f) upper BCTZ (450 – 800 m), (g) and (h) lower BCTZ (800 – 1200 m), (i) and (j) clathrate hydrate ice zone (1200 – 2000 m), and (k) and (l) outer ice (bubbly ice and clathrate hydrate ice).

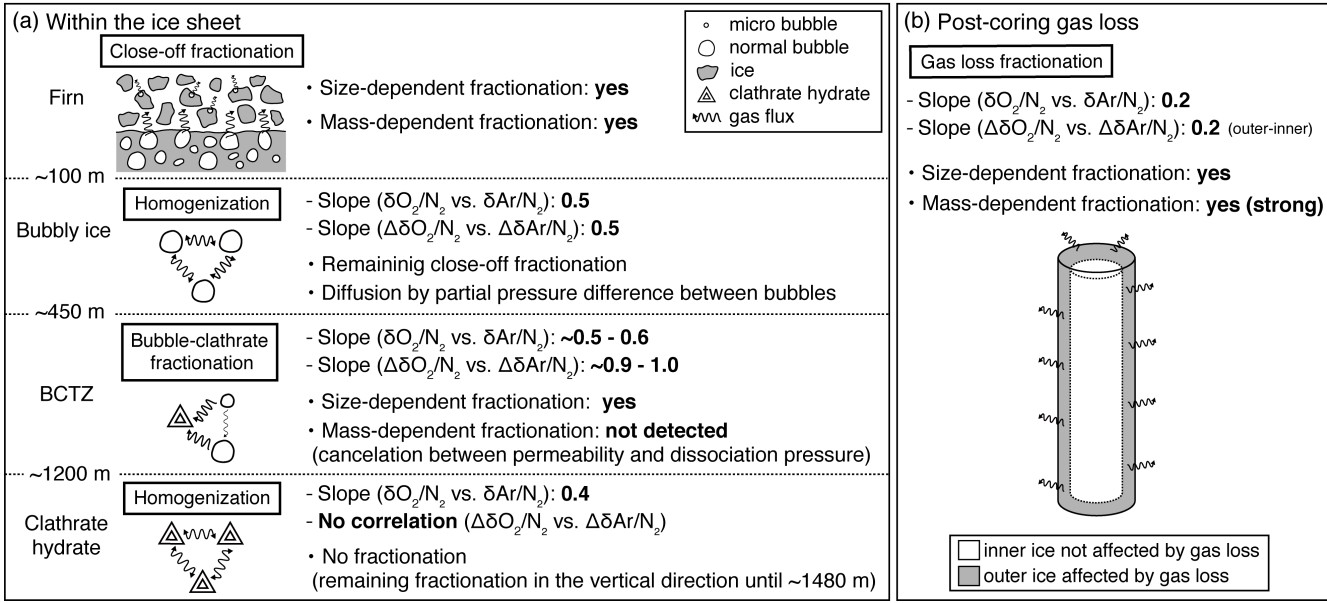

**Figure 11:** Schematic of fractionations of $O_2$ and Ar (a) within the ice sheet at Dome Fuji and (b) for the post-coring gas loss. The slopes
were taken by the bulk data and the pair difference of $\delta O_2/N_{2gravcorr}$ and $\delta Ar/N_{2gravcorr}$, respectively (see Fig. 10). The descriptions of "yes"
indicate the observed results (this study; Severinghaus and Battle, 2006; Huber et al., 2006; Battle et al., 2011).

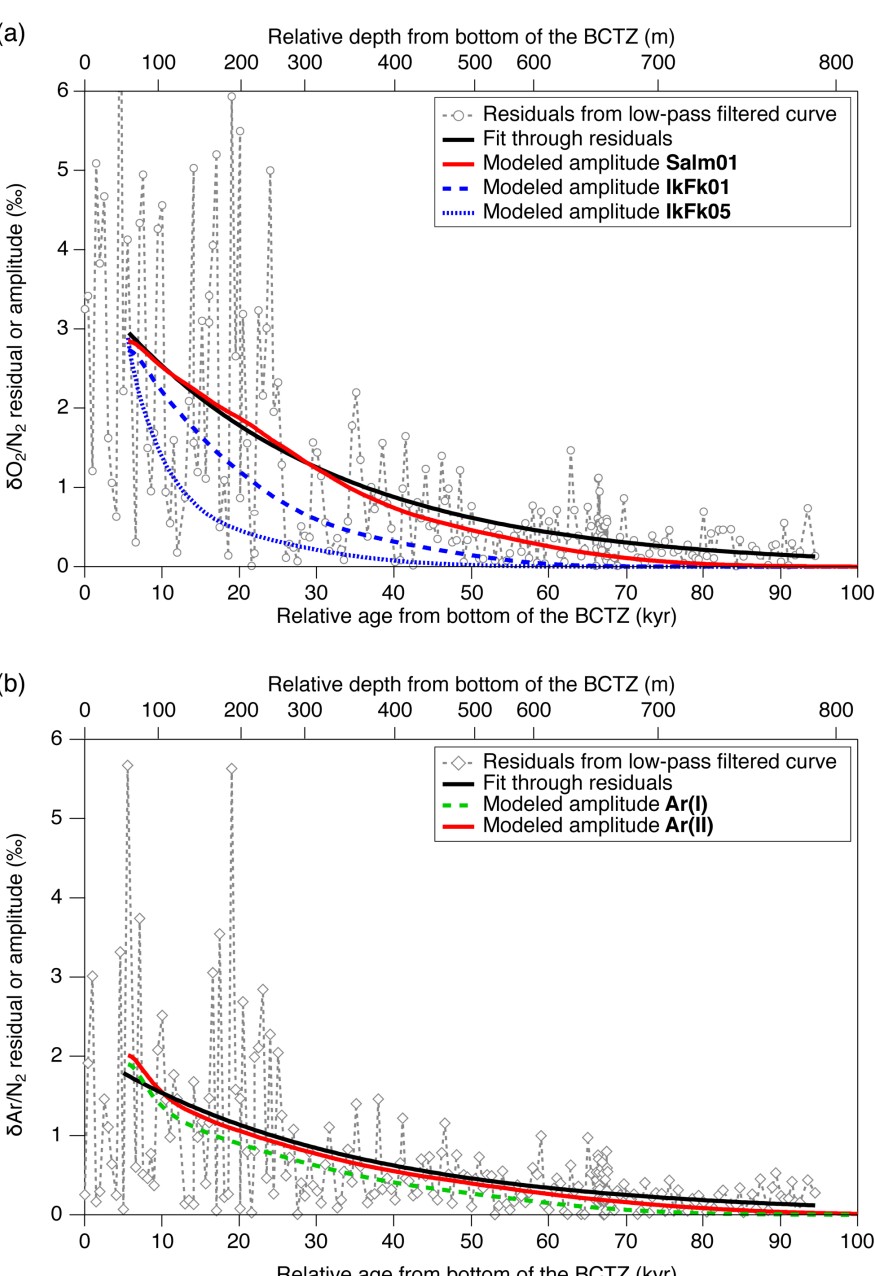

**Figure 12:** Diffusive smoothing of (a) $\delta O_2/N_2$ and (b) $\delta Ar/N_2$ below the BCTZ. Open markers with dashed lines (gray) are the residuals from the low-pass filtered data. Black lines are exponential fits through the residuals for > 5600 years (corresponding to the model's initial age). The model results (standard deviations of 11-cm resampled outputs) with the best permeation parameters are shown by solid red lines. Blue and green lines are the other model results.

# Tables

**Table 1:** Correlation coefficients between high-resolution gas data and ion concentrations within two 50-cm samples.

| | $\delta O_2/N_{2gravcorr}$ | $\delta Ar/N_{2gravcorr}$ | $Cl^-$ | $SO_4^{2-}$ | $NO_3^-$ | $F^-$ | $CH_3SO_3^-$ | $Na^+$ | $Mg^{2+}$ | $Ca^{2+}$ | $NH_4^+$ |
|---|---|---|---|---|---|---|---|---|---|---|---|
| **1258.5 – 1259.0 m** | | | | | | | | | | | |
| $\delta O_2/N_2$ | 1.00 | | | | | | | | | | |
| $\delta Ar/N_2$ | 0.75 | 1.00 | | | | | | | | | |
| $Cl^-$ | -0.18 | 0.07 | 1.00 | | | | | | | | |
| $SO_4^{2-}$ | 0.01 | 0.10 | 0.33 | 1.00 | | | | | | | |
| $NO_3^-$ | 0.37 | 0.20 | -0.49 | **-0.83** | 1.00 | | | | | | |
| $F^-$ | 0.47 | 0.37 | -0.37 | -0.46 | **0.78** | 1.00 | | | | | |
| $CH_3SO_3^-$ | **0.62** | **0.51** | -0.10 | -0.43 | **0.67** | 0.52 | 1.00 | | | | |
| $Na^+$ | **0.61** | **0.71** | 0.24 | 0.10 | 0.15 | 0.39 | **0.60** | 1.00 | | | |
| $Mg^{2+}$ | **0.72** | **0.74** | 0.21 | 0.16 | 0.26 | **0.58** | **0.60** | **0.82** | 1.00 | | |
| $Ca^{2+}$ | **(0.95)** | **(0.93)** | (-0.04) | **(0.77)** | (0.56) | (0.37) | **(0.83)** | **(0.75)** | **(0.78)** | (1.00) | |
| $NH_4^+$ | 0.00 | 0.11 | -0.54 | -0.03 | 0.56 | 0.45 | 0.30 | -0.02 | 0.05 | 0.14 | 1.00 |
| **1399.0 – 1399.5 m** | | | | | | | | | | | |
| $\delta O_2/N_2$ | 1.00 | | | | | | | | | | |
| $\delta Ar/N_2$ | 0.70 | 1.00 | | | | | | | | | |
| $Cl^-$ | 0.23 | 0.13 | 1.00 | | | | | | | | |
| $SO_4^{2-}$ | 0.43 | 0.30 | -0.42 | 1.00 | | | | | | | |
| $NO_3^-$ | -0.40 | -0.21 | 0.21 | **-0.83** | 1.00 | | | | | | |
| $F^-$ | -0.03 | -0.05 | **0.71** | **-0.52** | 0.40 | 1.00 | | | | | |
| $CH_3SO_3^-$ | -0.21 | 0.06 | 0.36 | -0.24 | 0.36 | 0.47 | 1.00 | | | | |
| $Na^+$ | **0.53** | **0.48** | 0.47 | **0.52** | -0.43 | 0.17 | 0.32 | 1.00 | | | |
| $Mg^{2+}$ | **0.54** | **0.43** | **0.52** | 0.48 | -0.42 | 0.26 | 0.29 | **0.96** | 1.00 | | |
| $Ca^{2+}$ | **0.52** | **0.44** | **0.55** | 0.12 | -0.20 | 0.44 | 0.06 | 0.54 | **0.72** | 1.00 | |
| $NH_4^+$ | -0.06 | 0.00 | -0.47 | **0.65** | **-0.71** | **-0.54** | -0.32 | 0.13 | 0.06 | -0.18 | 1.00 |

Bold numbers indicate significant correlations ($p < 0.05$). Numbers in brackets are calculated using the data from the lower 20 cm (because of the experimental problem in the $Ca^{2+}$ data).
