# Peer review of "Fractionation of $O_2/N_2$ and $Ar/N_2$ in the Antarctic ice sheet during bubble formation and bubble-clathrate hydrate transition from precise gas measurements of the Dome Fuji ice core"

_The Cryosphere, 2021_

## Author Comment (AC1)

Thank you for your valuable comments. Our responses and the changes we plan to make in the revised manuscript are explained below. Our replies are in blue, the changes we plan to make in red and reviewer comments are written in black italic letters.

*This paper reviews all of the non climatic mechanisms responsible for dO2/N2 fractionation in ice cores, and includes a compilation of most existing measurements, in addition to a large dataset of new measurement from the Dome F ice core, with a variety of sampling strategies.*
*It is an extremely useful and valuable paper, that clearly identifies optimal strategies for sampling ice cores to retrieve a valuable dO2/N2 signal. In particular, the authors did a great effort in cutting the ice in various ways to identify where and how dO2/N2 fractionation was happening, both between outer and inner ice samples, and with high resolution vertical sampling. Their new datasets shed new light on processes affecting gas loss in ice cores. The paper is very well written, the scientific quality of the analyses is excellent, and the figures are also very clear. Although I have a few important comments on the presentation of the results, I recommend its rapid publication.*

*Major comments:*
*1.      Section 4, called "Discussion" actually has results in it, and I am missing a "discussion" section that would include perspectives. I think it would be valuable to add a small "perspective" section, devoted to next steps: Do you think that everything is known about the different mechanisms of fractionation? What could/should be done next to improve either our understanding and isolation of the various processes, or improve the corrections? In particular, a summary on how your data could be picked up by ice physicists to test hypotheses about the mechanisms that you talk about in the paper would be an important addition, to make sure that your results are re-used.*

We will move the results of the diffusion model (L368 – 376) to the Results chapter.

For the perspective, we will add the following paragraph in the Conclusion chapter (and delete the last part of the first paragraph of Conclusion, which has some perspectives).

"The primary application of the $\delta O_2/N_2$ record has been the orbital tuning of the ice-core age scales. In the future, high-precision $\delta O_2/N_2$ and $\delta Ar/N_2$ data of the Dome Fuji core may be obtained with our technique for precise orbital tuning of the ice core. The high-precision data may also provide non-insolation signals on the gases and eventually be useful for reconstructing past atmospheric oxygen and argon concentrations.

More observational and theoretical works are still needed for advancing our understanding of the mechanisms of gas movements in different zones in the ice sheet. For example, the current precisions

of ice-core measurements of $\delta^{18}O$ of $O_2$ and $\delta^{40}Ar$ are insufficient for detecting mass-dependent fractionation during bubble close-off and bubble-clathrate transformation processes (note that the mass-dependent fractionation of $\delta^{18}O$ during bubble close-off was evidenced by the WAIS Divide firn-air data, Battle et al., 2011). Theoretical works including molecular dynamics simulations for different gases and ice conditions may shed light on the different relationships between $\delta O_2/N_2$ and $\delta Ar/N_2$ in different zones. Finally, our constraints on the permeation coefficients of the gases in ice might be useful for predicting the magnitude of diffusive smoothing of air composition in the ice sheet more than 1 million years old to be drilled in the future."

*2.      Related to my first comment, I am missing a summary at the end of the paper of the optimal sampling strategy. I think it would be great, either to add it to the conclusion, or to have a "sampling strategy" summary section, separated for each of the zones that you added. This section could summarize for each zone, the mechanisms controlling non gravitational d02/N2 signal, and how to mitigate these.*

The magnitude of the post-coring gas loss depends on the ice-core quality, sample size and dimensions, and storage temperature and period. The frequency and magnitude of the natural fractionation in the BCTZ might be related to the annual layer thickness and impurity concentrations. Therefore, an optimal sampling strategy for the Dome Fuji ice core should not be readily applicable to other ice cores. Instead, we will add our recommendations for finding the optimal sampling strategy for other cores. We will add the following section at the last part of Discussion.

**"4.5 Optimal storage and sampling strategy**
We discuss here the recommended practices for the storage and measurement of a newly drilled ice core based on our data. For long-term storage, it is more advantageous to have a larger ice-core cross-section and lower temperature. Based on our Dome Fuji data (~1 cm from the surface is affected by the gas loss at -50 ˚C after 20 years), a square cross-section of 3×3 cm seems sufficient in a -50 ˚C storage, for sampling a central part (cross-section of 1×1 cm or more) that is unaffected by the post-coring gas loss. The temperature of -50 ˚C was originally selected for inhibiting the clathrate hydrate dissociation due to the relaxation of ice matrix during long-term ice-core storage (Uchida et al., 1994). To obtain high-quality $\delta O_2/N_2$ and $\delta Ar/N_2$ data, it is recommended to test the real ice-core samples to find sufficient removal thickness. The removal thickness can be determined by examining the pair differences of $\delta O_2/N_2$ with different surface removal thicknesses (e.g., 5 and 8 mm or 3 and 5 mm; Oyabu et al., 2020), which should be within the measurement uncertainty. Five pairs or more for a given combination of removal thicknesses would be required to make the assessment.

The length of a sample is also an important ice-core-specific factor, especially for reasonably averaging the high scatters in the BCTZ. We speculate that the reasonable sample length in the BCTZ to obtain

a clear insolation signal may be more than 50 cm for the Dome Fuji ice core. We note that this length should be different for different ice cores because the thicknesses of the alternating layers of high and low clathrate concentrations should be different at different sites (Lüthi et al., 2010; Shackleton, 2019). To find a reasonable sample length for a core, it is advisable to continuously measure a ~1-m-long section with a ~2 cm resolution and examine various averaging lengths. The sample length should also be larger than one annual layer thickness to average out the seasonal layering (especially important for the cores with high accumulation rates)."

Uchida, T., Hondoh, T., Mae, S., Shoji, H., and Azuma, N.: Optimized Storage Condition of Deep Ice Core Samples from a View Point of Air-Hydrate Analysis, Mem. Nat. Inst. Polar Res., Special Issue, 49, 320-327, 1994.

*3.      For someone who is not close to the literature of O2/N2, it would be good to have, maybe in the introduction, or in your discussion section, a schematics of the fractionation mechanisms. You may have to have 3 or 4 of them, for each of your zones, and maybe one more for post-coring. Generally speaking, all of the mechanisms are discussed in a hand-wavy matter, and it's a bit difficult, through the reading of your paper, to understand what hypotheses about what mechanisms are actually testable or tested in your presentation.*

We will add a schematic in the revised manuscript.

[Figure]

**Figure caption:** Schematic of fractionations of $O_2$ and Ar (a) within the ice sheet at Dome Fuji and (b) for the post-coring gas loss. The slopes were taken by the bulk data and the pair difference of $\delta O_2/N_{2\text{gravcorr}}$ and $\delta Ar/N_{2\text{gravcorr}}$, respectively (see Fig. x). The descriptions of "yes" indicate the observed results (this study; Severinghaus and Battle, 2006; Huber et al., 2006; Battle et al., 2011).

*4.       In the conclusion, in addition to specific recommendations for sample handling and storage to avoid fractionation in the first place, I am missing an optimal strategy for correcting for gas loss. You find different slopes of O2/N2 vs Ar/N2 and for d18O vs O2/N2. I wonder if you could separate out the different mechanisms involved in "gas loss" fractionation, and provide several sequential correction strategies that could account for these differences, valid across different zones and different cores, to make it less heuristic.*

We did not propose a universal method of correction for all ice cores, because there are many determining factors on the post-coring gas loss (e.g., drilling quality, storage condition just after coring, and size, dimensions, temperature and period during the storage, etc.), which cannot be separately and quantitatively evaluated. Thus, we think it is sufficient to propose the method for excluding the gas-loss fractionation from the ice-core dataset, as we describe in the original manuscript.

For reference, empirical correction methods for $\delta O_2/N_2$ and $\delta^{18}O$ have been successfully established from fractionated datasets (e.g., if there is a clear relationship between $\delta O_2/N_2$ and the storage period, or between $\delta^{18}O$ and $\delta O_2/N_2$) (Kawamura et al., 2007; Severinghaus et al., 2009).

*Detailed comments:*

*1.       You don't explain clearly what you mean in the figures by dO2/N2_grav. It's only explained in Figure 8, but you use it already in Fig 4. You should briefly explain in Section 2 how (and why) you correct for gravitational fractionation, and introduce your notation.*

We will add the following sentences and equations in the third paragraph of section 2.2, and denote the gravitationally corrected ratios by adding subscripts "gravcorr" ($\delta O_2/N_{2gravcorr}$, $\delta Ar/N_{2gravcorr}$ and $\delta^{18}O_{gravcorr}$).

"The $\delta O_2/N_2$, $\delta Ar/N_2$ and $\delta^{18}O$ values relative to the modern atmosphere were corrected for the gravitational enrichment in firn, which is nearly proportional to the mass difference between the gas pairs (Craig et al., 1988). The gravitational correction can be estimated from $\delta^{15}N$ of the same sample (Sowers et al., 1989):

$$\delta_{gravcorr} = \delta - \Delta m \times \delta^{15}N \quad (1),$$

where $\delta_{gravcorr}$ is gravitationally corrected value, $\delta$ is measured value, and $\Delta m$ is the mass difference (4 for $\delta O_2/N_2$, 12 for $\delta Ar/N_2$, and 2 for $\delta^{18}O$)."

*2.       The section on your diffusion model is a bit difficult to understand. In the main text, you should start by saying what you want to do with this model, what you want to test. Do you want to validate the effective diffusivity? Do you want to predict how much diffusion there will be in newly drilled deep ice cores, to inform the sampling strategy?*

We aimed at testing with the model whether the observed homogeneity of the gas composition below BCTZ is quantitatively consistent with molecular diffusion in the ice sheet, with the independently proposed permeabilities of $N_2$, $O_2$ and Ar by previous authors. The prediction of gas diffusion for future ice corings is beyond the scope of this study, because it requires more advanced model setups (for significantly thinned ice near bedrock) and evaluation of temperature dependence of diffusivity.

*Then you might consider including the main equation, it does not take too much space, but helps the reader. It's maybe more useful than the Argon diffusivity (which is also described in the appendix), since O2 is the focus of your paper.*

We will add the main equation and a few associated equations that would help the readers.

- $\frac{\partial C_m^h}{\partial t} = \frac{\partial}{\partial z}\left(D_m \frac{\partial C_m^h}{\partial z}\right)$ (diffusion equation)

- $C_m^h = S_m P_m^d X_m$ (concentration of *m*-molecule ($m = N_2$, $O_2$ or Ar) dissolved in ice in equilibrium with clathrate hydrate)

- $log P_m^d = a_m - \frac{b_m}{T}$ (dissociation pressure)

*Finally, you need to describe better what you use for inputs, what you use for outputs, what are the tunable versus known parameters. In the main text, you need to have a sentence or 2 that have enough details that the reader understands what you are trying to do, and in the appendix, you need to add more details. As it is now, there is not enough information to reproduce your results. Can you describe more the model set up? How deep, the discretisation scheme you used, how long you ran it for, the input that you used, the outputs that came out.*

According to the comment, we will modify section 2.4 and Appendix A as follows (original sentences are in black and our changes are in red) to better describe the model.

(section 2.4)

[revised manuscript text omitted]

where $\Delta t$ is the time step ($10^6$ s $\approx$ 11.6 days).

*3.     Regarding the input of the diffusion model, have you considered using a tuned version of the Ca record, since the O2/N2 seems well correlated with Ca, rather than reproducing a shallow segment of data? There is a funny offset in your figures, that I presume cannot be interpreted, but it would be interesting to see if we could use Ca2+ your diffusion model as a predictor (and then, maybe, as a correction factor).*

Following the suggestion, we conducted simulations with the scaled $Ca^{2+}$ data as the model's initial states. As we have two depths with detailed chemistry data along with the gas data (1258 m and 1399 m), we used the 1258 m data to establish the Ca-$\delta O_2/N_2$ and Ca-$\delta Ar/N_2$ relationship. We converted the 1399-m Ca record to $\delta O_2/N_2$ and $\delta Ar/N_2$, and used them as the initial state at 1258 m (we also considered the ice thinning). The model results with the permeability of Salm01 and Ar(II) are shown below. For both $\delta O_2/N_2$ and $\delta Ar/N_2$, the model results agree with the data rather well in terms of the number of wiggles and their positions, as the reviewer speculated. However, the amplitudes are both overestimated and underestimated by the model (e.g., the model overestimates the amplitude at 1399.1 m and underestimates at 1399.25 m for both $\delta O_2/N_2$ and $\delta Ar/N_2$). As the initial states of $\delta O_2/N_2$ and $\delta Ar/N_2$ would not be perfectly reconstructed by scaling the Ca record, some mismatches would be expected (some other impurities such as Na and Mg might also be related because they are also highly correlated with $\delta O_2/N_2$).

This modeling exercise has the advantage that it doesn't have the arbitrariness of the relative phases in comparing the model results and data. However, the application of this alternative input is limited to only one case (1399 m) with a few wiggles. Thus, we would rather refrain from adding it to the manuscript.

[Figure]

**Figure caption:** Comparison of the diffusion model outputs for (a) $\delta O_2/N_2$ and (b) $\delta Ar/N_2$ with Salm01 and Ar(II) permeation parameters (red lines) and high-resolution data (blue and green dotted lines) at 1399 m. For this simulation, the initial states were constructed from the measured $[Ca^{2+}]$ profile with linear equations established from the data at 1258 m; $\delta O_2/N_2 = 2.8033\ [Ca^{2+}] - 33.55$ and $\delta Ar/N_2 = 5.9868\ [Ca^{2+}] - 61.467$.

*4.      Raman spectroscopy is not described at all (line 186). You could add a sentence or 2 describing how this technique does, what it shows. I expect your readership will have a lot of ice core scientist who may not be too familiar with it. Just 1-2 sentences would help them understand (even if of course, they could read the cited paper).*

We will add the following descriptions (original sentences are in black and our changes are in red).
"In the lowermost part of this range (below 720 m), individual clathrate hydrates with extremely enriched $\delta O_2/N_2$ (> ~1000 ‰) are found by Raman spectroscopy (Fig. 3h) (Ikeda-Fukazawa et al., 2001), in which laser light is focused on individual bubbles or clathrate hydrates, and the shift of wavelength and intensity of scattered light (Raman spectra) are measured for quantifying $O_2$ and $N_2$. The compositional ratio of $O_2$ and $N_2$ is assumed to be equal to the ratio of their Raman peak intensities."

*Line 270 : precise before/after gravitational correction.*

In the revised manuscript, we will denote them as $\delta O_2/N_{2\,gravcorr}$ and $\delta Ar/N_{2\,gravcorr}$ to indicate they are after gravitational correction.

*Line 310 : You say that highly fractionated bubbles and clathrates are stratified in mm scale samples. Then, if you average over a certain depth, do you retrieve a better signal? Later on, you say 50cm, but here, you could go into a bit more detail, and justify this number. I also wonder if averaging is enough, or if, in addition to this layering, you have selective fractionation (perhaps post coring, or diffusion) that creates a bias.*

The reviewer is right that if we average over a certain length, we should be able to retrieve a better signal. We agree that we can explain the idea and justify the acceptable length here.

The justification of the 50 cm is as follows. On the one hand, the acceptable scatter of $\delta O_2/N_2$ around the orbital-scale low-pass filtered curve is ~2 ‰ (one standard deviation), as seen for the depths just below the BCTZ (1200 – 1480 m) where the similarity to local summer insolation is observed. Also, the dataset of Kawamura et al. (2007) has scatters of 1.2 – 1.3 ‰ around the orbital-scale $\delta O_2/N_2$ variations. On the other hand, if we add a thin (e.g., 1-mm-thick) layer with extremely fractionated $\delta O_2/N_2$ of +1000 ‰ at one end of a 50-cm-long sample, it creates an anomaly of +2 ‰ (1000 ‰ × 0.1 cm / 50 cm) to the original $\delta O_2/N_2$ of the 50-cm sample. The exact thickness and $\delta O_2/N_2$ of anomalous layers can vary in real situations, but the above assumption is quite extreme (assuming all air inclusions in the thin layer has the maximum $\delta O_2/N_2$ observed by Raman spectroscopy), thus we expect that the length of 50 cm is sufficient for "diluting" the effect of anomalous layers.

The reviewer is also correct that the averaging would only work if we can eliminate the selective fractionations such as by post-coring gas loss. As we indeed excluded the gas-loss-fractionated outer ice from our dataset, we believe that the averaging is enough with our data. We will emphasize in the revision that the removal of gas-loss fractionated outer ice is a prerequisite.

We will add the following sentences at the end of the second paragraph of section 4.2.

"……… From these observations, we suggest that the highly fractionated bubbles and clathrate hydrates may be stratified in mm-scale layers, and that the scatters in our dataset may be produced by random inclusion of such fractionated layers at the top and/or bottom of the ice samples. For example, if a sample coincidentally includes a thin (e.g., 1-mm-thick) layer with $\delta O_2/N_2$ of +1000 ‰ at the top or bottom of a 10-cm-long ice, an anomaly of ~10 ‰ from the average $\delta O_2/N_2$ (excluding the anomalous layer) should result. We indeed observe the residual $\delta O_2/N_2$ of up to ~10 ‰ in the lower BCTZ around the orbital-scale fitting curve. Thus, by simply analyzing longer samples, the scatters created by the thin anomalous layers should be reduced. We suggest that a sufficient sample length to reduce the scatter to an acceptable level is ~50 cm, which would produce anomalies of up to ~2 ‰. With this noise level in the $\delta O_2/N_2$ data, the insolation signal should be reconstructed in the BCTZ, as seen in the somewhat scattered depths just below the BCTZ (1200 – 1480 m). We also emphasize that

the removal of the gas-loss fractionated outer ice is a prerequisite for the practice of averaging longer samples, for better reconstruction of average $\delta O_2/N_2$ in the ice sheet."

*Line 325: lower dissociation pressure should produce a steeper Ar partial pressure of gradient from bubbles to clathrate. Why? Can you explain a bit more ? (it's maybe obvious to you, but not to me)*

The description in the original manuscript was too brief and imprecise, thus we modify this sentence as follows. The basic idea is that the gas flux from bubbles to clathrates depends on permeation coefficient and dissociation pressure (Salamatin et al., 2001), with higher permeation coefficient and lower dissociation pressure leading to larger flux. For the case of $O_2$ and Ar, the former has a *higher* permeation coefficient and *higher* dissociation pressure, thus their effects on the flux should cancel to each other.

"For the BCTZ, the mass fluxes of gases from bubbles to clathrates through ice may depend on permeation coefficient and dissociation pressure (Eq. 8 in Salamatin et al., 2001), with larger permeation coefficient and lower dissociation pressure leading to larger flux. Thus, for the case of Ar and $O_2$, the lower permeation coefficient of Ar than that of $O_2$ ($2\times10^{-20}$ and $3\times10^{-20}$ $m^2$ $s^{-1}$ $MPa^{-1}$ at 240K, respectively) may be counteracted by the lower dissociation pressure of Ar than $O_2$ (3.5 and 4.9 Mpa at 240K, respectively), to result in similar relative fractionation between bubbles and clathrates with respect to $N_2$. This hypothesis may explain …"

*Paragraph near line 355 : Add that nucleation increases dO2/N2. If you average over two annual cycles of Ca, do you get back a good value or is it also biased? Is there a selective loss for some layers?*

We will add that nucleation increases $\delta O_2/N_2$ and $\delta Ar/N_2$ as the following.
It appears that the averaging over two cycles of Ca, which seems to roughly correspond to red lines (25-cm average) in Fig. 6, may be sufficient for obtaining a good value, with slight biases in some cases (1292 m in Fig. 6).

"Thus, the high-micro-inclusion layers may create early clathrate nucleation, which attract $O_2$ and Ar from air bubbles in the adjacent layers with fewer micro-inclusions, and increase their $\delta O_2/N_2$ and $\delta Ar/N_2$."

*Could the correlation to Ca also be due to different bubble sizes for different densification rates in the firn, like in Freitag's papers?*

We do not think that the correlation to Ca is related to the densification rate of firn. If Ca or some impurities (see below for a supplementary note) make smaller pores and enhance firn densification, the high-impurity layers close off earlier than the low-impurity layers in the close-off region in firn. This would promote bubble compression and thus $O_2$ depletion in the high-impurity layers. Thus, the correlation between $Ca^{2+}$ and $\delta O_2/N_2$ from this mechanism is expected to be negative, which is opposite to the observation.

As a supplementary note, on the link between impurities and firn densification rate, Fujita et al. (2014, 2016) suggested that, based on the observations of firn at Dome Fuji in Antarctica and NEEM in Greenland, the actual active agent for the layered firn densification rates are ions such as $Cl^-$, $F^-$ and $NH_4^+$, and the correlation with $Ca^{2+}$ is possibly superficial (as Ca is a major element to form salts). In our high-resolution ion data, the correlation between the ions ($Cl^-$, $F^-$ and $NH_4^+$) and $Ca^{2+}$ are generally low because they are mobile in ice so that they are lost during firn densification or smoothed in ice. In any case, the layered firn densification rate is probably irrelevant to the observed correlation between $Ca^{2+}$ and $\delta O_2/N_2$, as explained above.

Fujita, S. et al. (2014). Densification of layered firn of the ice sheet at NEEM, Greenland. *Journal of Glaciology*, *60*, 905–921. http://doi.org/10.3189/2014JoG14J006
Fujita, S et al. (2016). Densification of layered firn in the ice sheet at Dome Fuji, Antarctica. *Journal of Glaciology*, *62*(231), 103–123. http://doi.org/10.1017/jog.2016.16.

*Line 368: Here, it's difficult to understand the hypothesis, the inputs and the results. What mechanisms are you taking into account? Do you start with very high bubble/clathrate layered concentrations?*

We test whether the decreasing scatters below BCTZ are due to diffusive homogenization of layered displacement of gas molecules originally created in the BCTZ (thus not due to disturbance of insolation signal on the gas fractionation at the firn-ice transition in the past). We also test different permeabilities proposed by several studies. The only mechanism in the model is the molecular diffusion through the ice lattice driven by the concentration gradient of dissolved gas in the ice, which is in equilibrium with clathrate hydrates. The initial conditions of the model are the actual $\delta O_2/N_2$ and $\delta Ar/N_2$ profiles at 1258 m, which show highly layered values, and we let the model homogenize the layerings. The evolutions of temperature and thinning are incorporated in the model to mimic the real ice sheet condition (the depth profiles of thinning and temperature are assumed to be constant through time). We obtain the smoothed $\delta O_2/N_2$ and $\delta Ar/N_2$ profiles according to the elapsed time, and compare the model results with the high-resolution continuous ice-core data to assess the model results with

different permeabilities. Please note that a similar simulation has been conducted by previous authors (Bereiter et al., 2014), but the study was limited without continuous high-resolution ice-core data to compare; thus, this is the first study that we can directly compare the model results with the detailed $\delta O_2/N_2$ and $\delta Ar/N_2$ in the ice sheet.

In the revised manuscript, we will modify the descriptions about the model as the following: (1) Methods and Appendix will include the aim and setups of the model, and (2) the description at lines 368 – 376 will be moved to Results, and (3) the following text (in red letters) will be added to Discussion (before line 378 in the original manuscript). (black letters are unchanged from the original manuscript)

"This study, for the first time, directly compares the diffusion model results with the detailed $\delta O_2/N_2$ and $\delta Ar/N_2$ in the ice sheet. The model could reproduce the smoothing of layered gas compositions as seen in the high-resolution continuous data (Fig. 10a and 11b). Also, the relationships between $\delta Ar/N_2$ and $\delta O_2/N_2$ in different zones (bubbles, BCTZ and clathrates) are similar to each other (slope of around 0.5). From these observations, we conclude that the large scatters just below the BCTZ originate in layered gas fractionations in the lower BCTZ, and that the subsequent decrease of scatters is due to diffusive homogenization. We thus disfavor the possibility that calls for a failure of the recording mechanism of insolation variations during the past firn-ice transition to generate the high scatters of $\delta O_2/N_2$ and $\delta Ar/N_2$ in and below the BCTZ.

We also analyze our data in a similar manner as the work by Bereiter et al. (2014). The standard deviations of the model results resampled at 11-cm intervals are compared with the residual $\delta O_2/N_2$ and $\delta Ar/N_2$ data from the low-pass filtered curves (Fig. 12). Exponential fitting curves through the residual data (black line) are in close agreements with the model results with the Salm01 and Ar(II) permeation parameters. On the other hand, the model results with the other parameters (IkFk01, IkFk05 and Ar(I)) show too rapid decrease of scatters in comparison with the data. Therefore, our datasets (both high-resolution and normal datasets) consistently support the Salm01 and Ar(II) permeation parameters at around 240 K (temperature at DF for the simulated depths).

From the Salm01 parameters, the rate of diffusive migration is on the order of 0.1 mm per 10 kyr (10-10 m s-1). Therefore, we favor the interpretation that the extreme scatters of $\delta O_2/N_2$ and $\delta Ar/N_2$ in the BCTZ in our datasets are caused by mm-scale inhomogeneity of the compositions of air inclusions combined with the finite sample length, rather than by cm-scale bulk migration of gas molecules. We also suggest that the original insolation signal on $\delta O_2/N_2$ and $\delta Ar/N_2$ in the BCTZ may be reconstructed by analyzing long ice samples (>50 cm) to average out the inhomogeneity (see Section 4.2)."

*Line 389 : explain where the 50cm nb comes from, maybe above.*

*Figure 7: add some vertical bars to help the reader.*

We added vertical bars.

[Figure]

*Figure 8 and 9 : it would be useful to put them on the same figure (except panel 1), to help the comparison.*

Differences in axes ranges of bulk data and those of pair differences are too large to show in the same figure. Thus, we combined Figure 8 and Figure 9 in the same figure as shown below. The ratio of ranges of horizontal and vertical axes is 1:1 for each panel for easy comparisons of the slopes, except for panels (a) and (b).

---

## Author Comment (AC2)

**Reply to referee2**

Thank you for your valuable comments. Our responses and the changes we plan to make in the revised manuscript are explained below. Our replies are in blue, the changes we plan to make are in red and reviewer comments are written in black italic letters.

*The article "Fractionation of O2/N2 and Ar/N2 in the Antarctic ice sheet during bubble formation and bubble-clathrate hydrate transition from precise gas measurements of the Dome Fuji ice core" by Ikumi Oyabu presents new δO2/N2 and δAr/N2 data, measured on Dome Fuji ice cores using the method of Oyabu et al. 2020. The authors are able to provide data from samples stored at low temperatures of -50°C in the freezer and show that under these conditions gas loss fractionation after coring is almost negligible. They also discuss their data in the context of a wide range of δO2/N2 and δAr/N2 measurements from other ice core sites and other measurement and storage strategies. They examine their data in four depth intervals attributed to different fractionation mechanisms (bubble ice, upper BCTZ, deep BCTZ and clathrate zone) through a simple regression analysis of δO2/N2 versus δAr/N2 and δO2/N2 versus δ18O-O2 to disentangle possible fractionation mechanisms (mass-independent/size-dependent vs. mass-dependent fractionation). Furthermore, the authors show that using a simple diffusion model to model permeation in conjunction with high-resolution data can explain the reduction in data variance due to diffusive smoothing in the clathrate hydrate zone.*

*The paper is well written and structured. I enjoyed reading this paper and look forward to its publication. Most of my "major" criticisms of this work have already been addressed by reviewer 1, and I am pleased to see how the authors have responded. In particular, the new schematic illustration about the different fractionation mechanisms will help the reader to understand the work better. There are only a few minor points to change, which I list below.*

*Minor points:*

*Line 15: Please avoid expressions like "high precision" or specify with numbers.*

We will remove "at high precision".

*Line 21: Yes, analysing long ice samples can help to average the data scatter later, but how long should these samples be? Please specify a number here.*

We will add the proposed length as the following.

"… and the insolation signal may be reconstructed by analyzing long ice samples (more than 50 cm for the Dome Fuji core)."

*Line 72/73: Please combine minus sign and number in the same line.*

Yes, we will check the format in the proof.

*Line 160ff: For the data shallower than 800 m, I do not see much agreement with the insolation data. The depth range is too short to support this statement. For the deeper depth range, I agree.*

We agree that the age range for the data shallower than 800 m is too short to robustly compare with the insolation curve as we could for the deeper part. However, for the shallower depths, we think we can identify the similarity of $\delta O_2/N_2$ and insolation curve in that both curves show two peaks at ~350 m (12 kyr) and ~700 m (32 kyr), and that the second peak (~700 m) is larger than the first one. The comparison of $\delta Ar/N_2$ with the insolation curve is even less robust perhaps because the signal-to-noise ratio of $\delta Ar/N_2$ is smaller than $\delta O_2/N_2$. We have also observed that $\delta Ar/N_2$ has sometimes slightly different phasing with respect to $\delta O_2/N_2$ (in our ongoing measurements for older ages).
We will modify the text as follows.

"Variations in $\delta O_2/N_2$ and $\delta Ar/N_2$ for the depths ~~shallower than ~800 m and~~ deeper than ~1200 m have similarity with local summer insolation curve, while little similarity is found for 800 – 1200 m with extremely large scatters (Fig. 4). For the depths shallower than ~800 m, the comparisons between the gas records and insolation are less robust than for the deeper depths because of the short length (in terms of age) and small insolation amplitudes (small signal-to-noise ratio). Nevertheless, we find similarity between $\delta O_2/N_2$ and local summer insolation in that both curves show the two peaks at ~350 m (12 kyr BP) and ~700 m (32 kyr BP) and that the second peak (at ~700 m) is larger than the first one."

*Line 161: "We evaluate ... low-pass filtered curves": Please indicate here the cut-off period used and explain how the low-pass filtering was performed.*

We will explain the filter as follows. It is the same filter as used by Kawamura et al. (2007).

"We assess the scatters in the data by taking residuals of $\delta O_2/N_2$ and $\delta Ar/N_2$ from their low-pass filtered

curves (Fig. 3d and 3e). The low-pass filter (cut-off period: 16.7 kyr) and its usage are the same as in Kawamura et al. (2007). Briefly, we put the $\delta O_2/N_2$ and $\delta Ar/N_2$ data on the DFO-2006 time scale, lineally interpolated them at 0.1 kyr intervals, and applied the filter to extract their orbital-scale variations."

*Line 255/256: As already stated, the similarity of the bubble-ice data to the solar radiation curve is not robust in my opinion.*

We agree that the similarity is less robust than for the deeper depths, which may be mostly due to the (inevitable) short length for comparison as discussed above, thus we will weaken the statement and describe this part as a simple observation and consistency with the proposed mechanism (occurring in association with the bubble formation processes). We will modify the text as follows.

"The $\delta O_2/N_2$ and $\delta Ar/N_2$ data  below BCTZ show variations similar to the local summer insolation (Fig. 4). In addition,  we find the possible insolation signals in the bubbly ice zone and upper BCTZ (see 3.1) ,  as expected from the proposed link between the local summer insolation and close-off fractionation through the effects on the snow metamorphism (Bender, 2002; Fujita et al., 2009)."

---

## Referee Report (RR1)

The authors have followed the reviewer's recommendations closely. All major and minor comments have been adequately addressed and the authors have made some significant changes to the manuscript to further improve the paper. In my opinion, the paper is now ready for publication and I look forward to its publication.

I only found three minor language issues and a formatting problem that may need to be changed before the paper is finalized (depends on the editor's opinion).

Minor changes:

Lines 390ff:

"For example, if a sample coincidentally includes a thin (e.g., 1-mm-thick) layer with $\delta O_2/N_2$ of +1000 ‰ at the top or bottom of a 10-cm-long ice, an anomaly of ~10 ‰ from the average $\delta O_2/N_2$ (excluding the anomalous layer) should result."

I think that sentence is confusing.

Line 482ff:

"We thus disfavor the possibility that calls for a failure of the recording mechanism of insolation variations during the past firn-ice transition to generate the high scatters of $\delta O_2/N_2$ and $\delta Ar/N_2$ in and below the BCTZ."

I think that sentence is confusing. Please remove double negative phrase.

Line 594:

"More observational and theoretical works are still needed for advancing our understanding of the mechanisms of gas movements in different zones in the ice sheet."

Change "works are" to "work is"

Eq. A9 to A12: There is something wrong with the indices in the brackets, the brackets do not line up with the indices.

---

## Author Response (AR2)

**Author response to the review of Oyabu et al. "Fractionation of $O_2/N_2$ and $Ar/N_2$ in the Antarctic ice sheet during bubble formation and bubble-clathrate hydrate transition from precise gas measurements of the Dome Fuji ice core"**

We would like to thank Dr. Nanna B. Karlsson and the anonymous reviewer for comments to improve and clarify the manuscript. We revised the manuscript following the comments. Our replies are in blue, the changes we made in red and the editor and reviewer comments are written in *black italic letters*.

**Reply to the Editor**

*Line 90: "(corresponding to ~10 years)" - is this correct? Presumably the age span varies greatly with depth so the 10 years is not correct for the entire ice core.*

We changed to "(corresponding to ~3 to 30 years)" (Line 89).

*Line 600: "Finally, our constraints on the permeation coefficients of the gases in ice might be useful for predicting the magnitude of diffusive smoothing of air composition in the ice sheet more than 1 million years old to be drilled in the future."*
*Is there a word missing? This sentence is difficult to read, I suggest splitting it up and referencing the Oldest Ice effort, e.g., "Finally, the International Partnership for Ice Core Sciences regards the retrieval of an ice core containing ice older than 1 million years as highest priority (Fisher et al., 2013). Our constraints on the permeation coefficients of the gases in ice might be useful for predicting the magnitude of diffusive smoothing of air composition in such an ice core."*
*You could also add (if appropriate) a reference the manuscript by Tsutaki (in review https://tc.copernicus.org/preprints/tc-2021-266/) for a specific Dome Fuji reference.*

*Fisher et al., 2013: Clim. Past, 9, 2489–2505, 2013, www.clim-past.net/9/2489/2013/, doi:10.5194/cp-9-2489-2013*

We revised as the editor suggested (Line 520, added Fischer et al., 2013 and Tsutaki et al., in review).

In addition to the above corrections, we added information about data citation in the Data availability "(https://ads.nipr.ac.jp/dataset/A20210430-001; https://doi.org/10.17592/001.2021043001, Oyabu et al., 2021)" and added the corresponding reference in the reference list.
Oyabu, I., Kawamura, K., Kitamura, K., Hirabayashi, M., Aoki, S., Morimoto, S., and Nakazawa, T.: Dome Fuji ice core gas data (112 - 2001m, $^{15}N$, $O_2/N_2$, $Ar/N_2$), 1.00, Arctic Data archive System (ADS), Japan, http://doi.org/10.17592/001.2021043001, 2021.

**Reply to Referee #2**

*Lines 390ff:*
*"For example, if a sample coincidentally includes a thin (e.g., 1-mm-thick) layer with δO2/N2 of +1000 ‰ at the top or bottom of a 10-cm-long ice, an anomaly of ~10 ‰ from the average δO2/N2 (excluding the anomalous layer) should result."*
*I think that sentence is confusing.*

We revised the sentence as follows (Line 356):
For example, if a 100-mm-long ice sample coincidentally includes a 1-mm-thick anomalous layer with $\delta O_2/N_2$ of +1000 ‰, the $\delta O_2/N_2$ of the bulk sample should be elevated by ~10 ‰ relative to the value without the anomalous layer.

*Line 482ff:*
*"We thus disfavor the possibility that calls for a failure of the recording mechanism of insolation variations during the past firn-ice transition to generate the high scatters of δO2/N2 and δAr/N2 in and below the BCTZ."*
*I think that sentence is confusing. Please remove double negative phrase.*

We revised the sentence as follows (Line 430):
Thus, we favor the possibility that the recording mechanism of insolation variations at Dome Fuji was intact when the layers in the modern BCTZ and below were initially formed at the past firn-ice transition.

*Line 594:*
*"More observational and theoretical works are still needed for advancing our understanding of the mechanisms of gas movements in different zones in the ice sheet."*
*Change "works are" to "work is"*

Corrected (Line 515).

*Eq. A9 to A12: There is something wrong with the indices in the brackets, the brackets do not line up with the indices.*

It might be a problem with the software that displays the PDF file. On my laptop, it looks like the picture that I attached. I will carefully check the brackets in the proof.
* * *
**Discretization**

The model uses the central differencing scheme. The downward diffusive flux ($f_m$) of $m$-molecule per unit area at the top boundary of $i$-th box is the product of the diffusivity and concentration gradient:

$$f_{m(i)} = D_m \frac{c_{m(i-1)}^h - c_{m(i)}^h}{\Delta z \tau_r}. \qquad (A9)$$

where $\Delta z$ is initial box height (0.5 mm) and $\tau_r$ is relative thinning function (thinning function divided by the initial value at 1258 m). By substituting eq. 3 into eq. A9, $f_m$ is expressed as

$$f_{m(i)} = \frac{D_m S_m P_m^d}{\Delta z \tau_r} \left( X_{m(i-1)} - X_{m(i)} \right). \qquad (A10)$$

The net flux of $m$-molecule for $i$-th box is

$$F_{m(i)} = \frac{f_{m(i)} - f_{m(i+1)}}{\Delta z \tau_r}, \qquad (A11)$$

and the concentration change of $m$-molecule in total air content becomes

$$\Delta U_{m(i)} = F_{m(i)} \Delta t, \qquad (A12)$$

where $\Delta t$ is time step (~11.6 days).